# H3K4me2 distinguishes a distinct class of enhancers during the maternal-to-zygotic transition

Matthew D. Hurton, Jennifer M. Miller, Miler T. Lee *

Department of Biological Sciences, University of Pittsburgh, Pittsburgh, Pennsylvania, United States of America

* miler@pitt.edu

## Abstract

After egg fertilization, an initially silent embryonic genome is transcriptionally activated during the maternal-to-zygotic transition. In zebrafish, maternal vertebrate pluripotency factors Nanog, Pou5f3 (OCT4 homolog), and Sox19b (SOX2 homolog) (NPS) play essential roles in orchestrating embryonic genome activation, acting as "pioneers" that open condensed chromatin and mediate acquisition of activating histone modifications. However, some embryonic gene transcription still occurs in the absence of these factors, suggesting the existence of other mechanisms regulating genome activation. To identify chromatin signatures of these unknown pathways, we profiled the histone modification landscape of zebrafish embryos using CUT&RUN. Our regulatory map revealed two subclasses of enhancers distinguished by presence or absence of H3K4me2. Enhancers lacking H3K4me2 tend to require NPS factors for de novo activation, while enhancers bearing H3K4me2 are epigenetically bookmarked by DNA hypomethylation to recapitulate gamete activity in the embryo, independent of NPS pioneering. Thus, parallel enhancer activation pathways combine to induce transcriptional reprogramming to pluripotency in the early embryo.

## Introduction

In animals, embryonic development begins with a transcriptionally silent zygotic genome under the control of maternally deposited RNAs and proteins [1,2]. In fast-dividing embryos of taxa such as Drosophila, Xenopus, and zebrafish, embryonic chromatin transforms over the course of several cleavages during the maternal-to-zygotic transition (MZT), leading to transcriptional competence and zygotic (embryonic) genome activation (ZGA) in the blastula [3–6]. Genome activation is facilitated in part by maternally deposited transcription factors that bind gene-proximal promoters and gene-distal enhancers in the embryonic genome [6–13]. In zebrafish, maternal Nanog, Pou5f3, and Sox19b (NPS) – homologs of the mammalian pluripotency factors NANOG, OCT4, and SOX2 – are essential for regulating a large proportion

**Data availability statement:** Sequencing data are available in the Gene Expression Omnibus (GEO) under accession number GSE269795. Analysis scripts are available at github.com/MTLeeLab/zf-k4 and archived at OSF under DOI https://doi.org/10.17605/OSF.IO/AUMHR.

**Funding:** M.T.L was supported by NIH grant R35GM137973 and March of Dimes grant 5-FY16-307. This study was also supported in part by the University of Pittsburgh Center for Research Computing, RRID:SCR_022735, specifically through the H2P cluster under NSF award number OAC-2117681. The funders had no role in the study design, data collection and analysis, decision to publish, or preparation of the manuscript.

**Competing interests:** The authors have declared that no competing interests exist.

**Abbreviations:** h.p.f., hours post fertilization; mESCs, mouse embryonic stem cells; MZT, maternal-to-zygotic transition; PCA, principal component analysis; SVM, support vector machine; TSS, transcription start site; ZGA, zygotic genome activation.

of genome activation [14–16], thus mechanistically linking mammalian pluripotency induction and the zebrafish MZT.

NPS, like their mammalian counterparts, act as pioneer factors capable of binding DNA regulatory sequences in the context of condensed chromatin [5,16–20], which tends to occlude binding of non pioneers [21,22]. Binding induces increased chromatin accessibility, leading to the acquisition of activating histone post-translational modifications such as acetylation and H3 lysine 4 (H3K4) methylation, which are correlated with the onset of embryonic gene transcription [16,23]. However, a triple maternal-zygotic mutant for *nanog*, *pou5f3* and *sox19b* (MZ*nps*) still activates some genes, implicating other unknown mechanisms that act alongside of NPS to regulate genome activation [16].

Chromatin is dynamic in the early zebrafish embryo. During the first two hours post fertilization (h.p.f.), genomics assays suggest that embryonic chromatin has limited accessibility and mostly lacks histone modifications [5,18,24–26]. Subsequently, a minor wave of genome activation begins, focused on a small number of gene promoters including the tandemly repeated microRNA *mir-430* encoding locus [27,28]. Chromatin accessibility and activating histone modifications have started to emerge, increasing by 4 h.p.f. (sphere stage) to tens of thousands of accessible, highly histone-modified promoters and enhancers [5,18,24,25,29–31]. By this point, the major wave of genome activation is underway, involving transcription of >7,000 genes, some of which are de novo expressed in the embryo (strictly zygotic), but the majority of which were already represented in the embryonic transcriptome from the maternal contribution (maternal-zygotic) [14,32].

Many of these chromatin changes require NPS pioneering, but several studies also implicate differential DNA methylation as being instructive for genome activation [5,26,33–39]. Both gametes contribute selectively 5-methylcytosine modified DNA, though rather than establishing differential parent-of-origin imprinted patterns like mice, zebrafish embryonic genome methylation is largely reprogrammed to match the paternal pattern by 3 h.p.f., through enzymatic-mediated methylation at some loci and passive demethylation at others [34,38]. Promoters that acquire or sustain hypomethylation recruit "placeholder" nucleosomes, characterized by H3K4 monomethylation (H3K4me1) and the histone variant H2A.Z (H2AFV in zebrafish), which help maintain hypomethylation and chromatin accessibility [37]. Hypomethylation at distal regulatory regions is also associated with dynamic regulation, though so far such regions have been found to co-occur with repressive histone modifications like H3K27me3 and H2Aub and thus may represent poised enhancers with roles later in development [33,35].

These observations implicate a combinatorial regulatory code underlying genome activation that may be further elucidated with additional characterization of embryonic chromatin. There are >100 different histone modifications described thus far [40], the vast majority of which are understudied in any context let alone in embryos. Recent work in mouse embryonic stem cells (mESCs) demonstrates that acetylation of the histone H2B N-terminal tail (H2BNTac) is strongly characteristic of enhancers as compared to most promoters [41]. Additionally, although most of the focus in gene

regulation literature has been on modifications of histone tails, acetylation in the core globular domain of histone H3 has recently been associated with enhancers as well. H3K56ac was shown to co-occur with Oct4 binding in mESCs [42], while H3K122ac and H3K64ac appear to mark a set of active enhancers lacking H3K27ac enrichment [43]. To our knowledge, these marks have not previously been evaluated in zebrafish.

H3K4 methylation has already been extensively profiled, but the logic dictating methylation degree at regulatory elements – i.e., mono-, di-, or tri-methylation – still needs to be more fully elucidated [44]. Classically, H3K4me3 has been associated with active transcription and is found promoter-proximal in gene bodies, while H3K4me1 and to some extent H3K4me2 is more diagnostic of enhancers [45–48]. Some studies have also found H3K4me3 at enhancers in certain contexts [45,49–52]; however, a recent analysis of several widely used H3K4 methylation antibodies has revealed a high prevalence of cross reactivity, calling into question the extent to which specific methylation degrees can be conclusively deduced at different regulatory elements [53]. Indeed, using new SNAP-ChIP verified antibodies, only H3K4me1 and H3K4me2, but not H3K4me3, are observed at enhancers in K562 cells [53]. These results motivate the re-evaluation of H3K4 methylation status in other systems.

Here, we have mapped the genome-wide distribution of 10 different histone modifications in the early zebrafish embryo using Cleavage Under Targets and Release Using Nuclease (CUT&RUN), to capture signatures of differentially-regulated enhancers and promoters during genome activation. We observe that characteristic combinations of these histone modifications broadly separate putative enhancers and promoters, but we also find that H3K4me2 and not H3K4me3 specifically marks a subclass of active enhancers, distinguishing them from other enhancers bearing only H3K4me1. Both H3K4me1 and H3K4me2-marked enhancers can distally regulate gene transcription. However, H3K4me1 enhancers largely rely on NPS pioneering to gain activity, whereas H3K4me2 enhancer activation is correlated with DNA hypomethylation that reflects their prior activity in gametes. Our findings reveal that differential H3K4me2 can distinguish enhancer subtypes, and that parallel pathways for enhancer activation underlie embryonic genome activation, explaining how some genes can still be activated in the absence of NPS pluripotency factors.

## Results

### CUT&RUN effectively maps histone modifications in zebrafish blastulae

We adapted and optimized CUT&RUN to zebrafish blastulae as a low-input alternative to conventional chromatin immunoprecipitation sequencing (ChIP-seq) [54–56]. We profiled embryos at the onset of dome stage – 4 to 4.3 h.p.f., the tail end of the major wave of genome activation (Fig 1A) – to assay the histone tail acetylation modifications H3K9ac, H3K27ac, H4K16ac, and H2BK16ac (an example of H2BNTac); the non-tail H3K56ac, H3K64ac, and H3K122ac modifications of the H3 histone globular core; and H3K4me1, 2, and 3 using SNAP-ChIP verified antibodies to precisely distinguish between methylation degrees (Fig 1B and S1 Table). Only 10 embryos per sample (approximately 70,000 cells [57]) were required to generate robust CUT&RUN libraries. We centered our analyses on genomic intervals flanking accessible chromatin as determined by ATAC-seq from two previously published studies [5,18] ($N$=48,395 open-chromatin regions), many of which likely represent active gene regulatory elements in the embryo (Fig 1B). To identify correlated histone mark enrichment patterns across the regions, we performed a principal component analysis (PCA) on CUT&RUN coverage across all samples (Figs 1B and S1A–S1C), and again pooling CUT&RUN replicates per mark (Figs 1C–1E and S1D–S1F). The first two principal components for both analyses broadly separate promoters – defined as open regions overlapping Ensembl, RefSeq, and UMMS [58] annotated transcription start sites (TSS) – and putative enhancers at least 2 kb from any TSS (Fig 1C and 1D and S2 Table).

However, some annotated enhancers cluster with the promoters and vice versa, indicating that these regions have histone modification patterns that resemble the other category (Fig 1E). Inspection of the PCA loadings revealed that H3K4 methylation strongly contributed to the first three principal components (S1C and S1F Fig). Focusing on regions marked by H3K4me1, when visualized in CUT&RUN coverage heatmaps, the promoters clustering with enhancers simply appeared to be less active compared to the other promoters, with weak acetylation, H3K4me2, and H3K4me3 (Figs 1F,

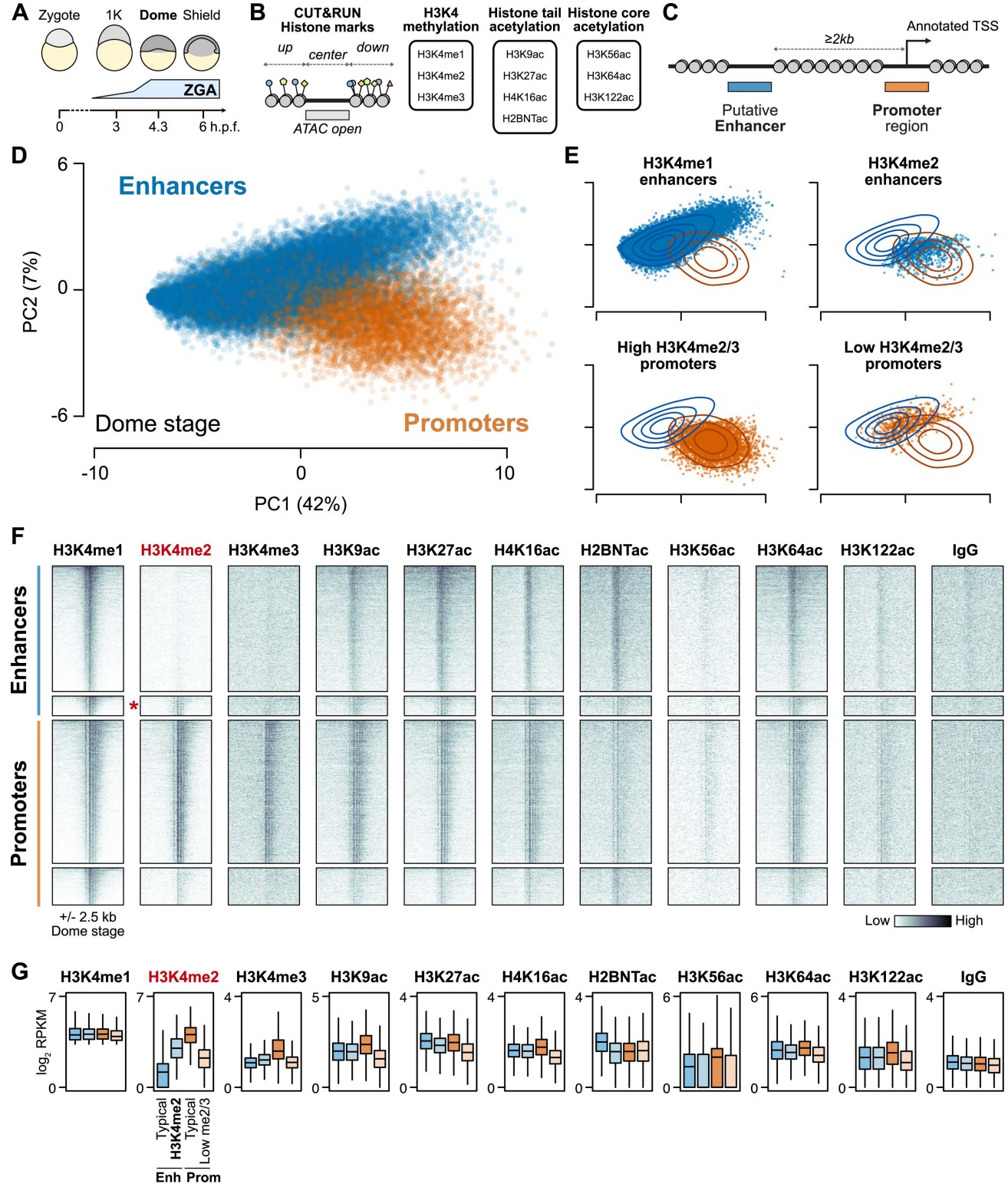

**Fig 1. Histone modifications distinguish regulatory elements during the maternal-to-zygotic transition. (A)** Schematic of early zebrafish embryogenesis spanning the 1-cell zygote, 1K-cell, dome, and shield stages, showing the timing of zygotic genome activation (ZGA). h.p.f. = hours post fertilization. **(B)** CUT&RUN read coverage was measured on open chromatin regions defined by ATAC-seq and adjacent 500-bp upstream and downstream

regions for 10 histone modifications. **(C)** Open chromatin regions were classified as TSS-overlapping promoters or TSS-distal putative enhancers. **(D)** Biplot of the first two principal components (PCs) of a PCA performed on replicate-pooled dome-stage histone modification coverage on open chromatin regions. Points are labeled blue for enhancers, orange for promoters, as defined in **(C)**. Percent of total variance explained per PC in parentheses. The data underlying this figure can be found in S1 Data. **(E)** PCA biplots separated according to support vector machine (SVM) classification on the first three PCs. "Typical" enhancers and promoters where the SVM classification matched the labels are plotted on the left panels, while regions where SVM classification disagreed with labels are plotted on right panels. Contour lines representing the density of enhancer (blue) and promoter (orange) points in the full PCA plot in **(D)** are overlaid. The four groups are named after their H3K4 methylation differences. The data underlying this figure can be found in S1 Data. **(F)** Heatmaps of replicate-pooled CUT&RUN coverage centered on a subset of H3K4me1-marked regions from each of the four groups defined in **(E)**. Individual replicates are shown in S2A Fig. Top to bottom, $N = 4{,}128$ typical "H3K4me1 enhancers," 644 "H3K4me2 enhancers" (marked with a red asterisk), 4,707 typical "High H3K4me2/3" promoters, and 1,224 "Low H3K4me2/3 promoters." **(G)** Boxplots summarizing the coverage observed in **(F)**. Boxes are first through third quartiles, center bar median, whiskers extend to 1.5× the interquartile range, outliers are not shown. H3K4me1 was used to select the regions, so differences between groups are expected to be minimal. For each of the remaining marks, significant differences were assessed by individual Kruskal–Wallis tests (H3K4me2, H3K4me3, H3K9ac, H3K27ac, H4K16ac, H2BNTac: $P < 1 \times 10^{-100}$; H3K56ac, H3K26ac, H3K122ac: $P < 1 \times 10^{-30}$). The data underlying this figure can be found in S1 Data. RPKM, reads per kilobase per million.

1G and S2A). Differential transcriptional activity is also supported by nascent RNA-seq at genome activation (S2B Fig). By contrast, the enhancers clustering with promoters had comparably strong acetylation to the other enhancers, but were additionally marked by H3K4me2, whereas most enhancers only had H3K4me1; we call these "H3K4me2 enhancers" and "H3K4me1 enhancers," respectively (Figs 1F, 1G and S2A).

Inspecting the other marks, we found that H3K4me3 was minimal in both enhancer classes (Figs 1F, 1G and S2A), consistent with the recent re-evaluation of H3K4 methylation degree at enhancers [53]. H2BNTac enrichment strongly contributes to the second principal component, anti-correlated with H3K4me2, and accordingly we observe stronger H2BNTac enrichment in the H3K4me1 enhancers (Figs 1F, 1G and S2A). Finally, we found that the core globular acetylation marks H3K56ac, H3K64ac, and H3K122ac, which have been linked to unique enhancer subtypes in other systems [42,43], do not seem to differentiate our two enhancer classes (Figs 1F, 1G and S2A). H3K56ac does contribute to the fourth principal component, but we did not discover any new enhancer subclasses in zebrafish induced by this or the other core acetylation marks (S2C and S2D Fig). Moving forward, we focused on further characterizing the strong dichotomy of H3Kme2-marked versus non H3K4me2-marked putative enhancers.

## H3K4me2-marked distal regions are likely a distinct class of bona fide enhancers

We first considered whether H3K4me2 might not be specific to the subset of enhancers at dome stage, but may instead be a temporally variable property of all enhancers. We performed additional CUT&RUN experiments at an earlier and later time point – 1K-cell stage (3 h.p.f.), just prior to the onset of the major wave of genome activation, and shield stage (6 h.p.f.), during gastrulation. We found that H3K4 methylation is overall weak at 1K-cell stage, with no evidence for H3K4me2 at any putative enhancer (Figs 2A and S3A). Using an alternate H3K4me1 antibody (Active Motif) and acknowledging the possibility of cross-reactivity, we do observe some enhancer H3K4 methylation enrichment, but this was largely restricted to the eventual dome-stage H3K4me2 enhancers (Figs 2A and S3A). At shield stage, the H3K4me2 presence/absence patterns observed at dome stage are largely preserved (Figs 2A and S3B). A principal component analysis of CUT&RUN coverage over time further supports the distinction between H3K4me1 enhancers and H3K4me2 enhancers, and indeed suggests different activation trajectories for the two enhancer classes (S3C–S3E Fig). So, it is unlikely that H3K4me2 is a generic property of all enhancers.

We next considered whether the putative H3K4me2 enhancers may in fact be unannotated gene promoters. H3K4me2 enhancers do not subsequently gain H3K4me3 (Figs 2A and S3B) nor do they specifically co-occur with repressive marks in previously published datasets for H3K27me3 [30,31], H3K9me3 [59], and H2Aub [33] (S4A Fig), suggesting that these regions are not poised promoters. Additionally, we queried two existing timecourse RNA-seq datasets [60,61] looking for evidence of gene-specific transcription but found only approximately 7% of H3K4me2 enhancers with any evidence for

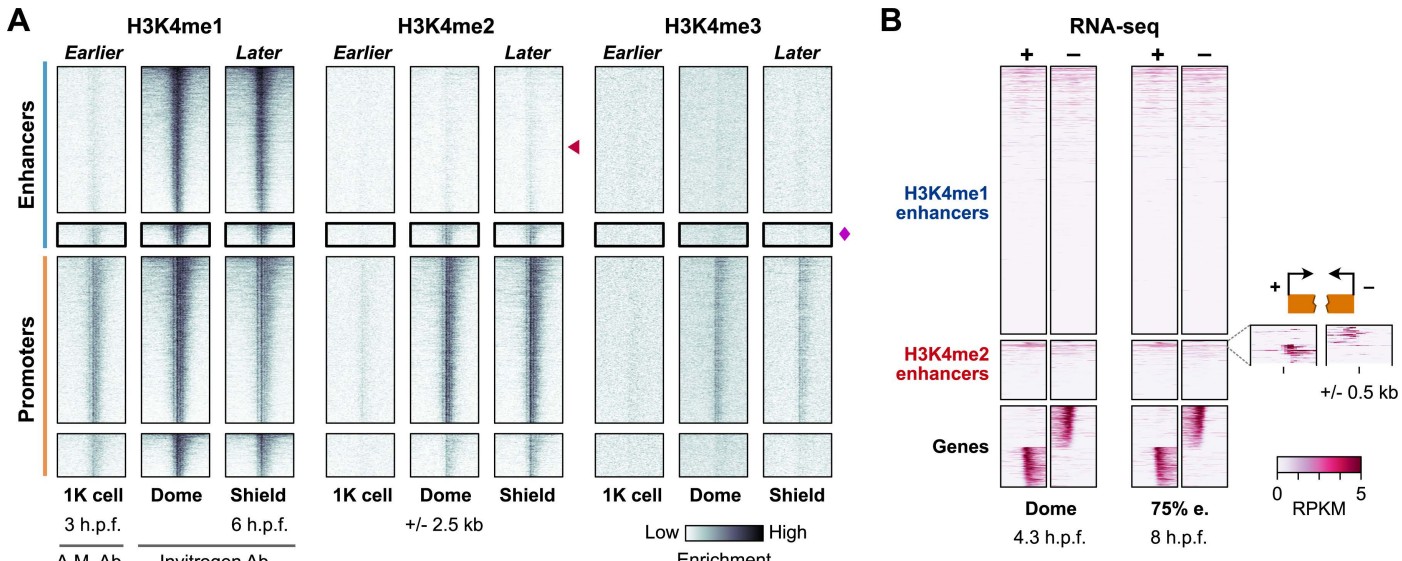

**Fig 2. Genomic profiles over time support a stable subset of H3K4me2-marked enhancers. (A)** Time course of replicate-pooled CUT&RUN coverage for the regions defined in Fig 1. Individual replicates are shown in S3A Fig. Red triangle points to the typical enhancers, which lack H3K4me2 coverage, magenta diamond marks the promoter-like enhancers, which do not gain H3K4me3. For 1K-cell stage, CUT&RUN for an Active Motif H3K4me1 antibody is shown, which yielded stronger signal than the Invitrogen antibody that was used for the other stages (see S2A Fig). **(B)** Heatmaps of strand-separated RNA-seq coverage centered on the H3K4me1 enhancers and H3K4me2 enhancers, with a subset of gene TSSs shown below to illustrate the expected pattern of unidirectional (−) strand read coverage extending upstream for (−) strand genes and (+) strand coverage extending downstream for (+) strand genes. A zoomed view of coverage at 75% epiboly stage (75% e.) over the top-covered H3K4me2 enhancers is shown to the right. h.p.f. = hours post fertilization, RPKM, reads per kilobase per million.

directional, stable transcripts (Figs 2B and S4B). Although the RNA-seq signal was weak, we removed these regions from subsequent analysis. After refining enhancer categories using dome-stage CUT&RUN enrichment thresholds along with the SVM classification, we focused subsequent analyses on 3,268 putative H3K4me1 enhancers (≥2-fold H3K4me1, ≥ 1.5-fold H3K27ac, and <1.25-fold H3K4me2 enrichment over IgG) and 681 putative H3K4me2 enhancers (≥2-fold H3K4me2 and ≥1.5-fold H3K27ac enrichment over IgG).

To assess the capacity for H3K4me2 enhancers to distally activate gene transcription, we designed and constructed reporter plasmids, cloning 29 putative regulatory elements each upstream of an mCherry open-reading frame with a minimal β-globin promoter (Fig 3A and S3 Table). Independent promoter activity is detected by divergent mTagBFP2 and EGFP open-reading frames (Figs 3A and S5A). We performed transient expression assays by injecting plasmid into 1-cell embryos and visualizing fluorescence at 6 h.p.f. to allow time for fluorophore transcription, translation, and maturation. Ten H3K4me2 enhancers and 10 H3K4me1 enhancers drove mosaic mCherry expression (likely due to injection variability) of varying intensity, demonstrating their enhancer capability (Figs 3B–3F and S5A–S5C). We additionally observed some mostly weak GFP or BFP expression for three H3K4me2 and three H3K4me1 reporters, suggesting some dual enhancer-promoter functionality (S5 Fig and S3 Table). Comparably-sized promoter sequences were generally insufficient to drive robust reporter expression (S5E–S5F Fig), with the exception of an *arhgap18* promoter element, which showed strong enhancer activity (S5E Fig). The *arhgap18* gene is not activated until well after the MZT, at 1–4 somites stage (10.33 h.p.f.) (S5G Fig), raising the possibility that its promoter could instead serve an alternate role as an enhancer during early development. Together our reporter assays support the existence of two distinct enhancer classes in the early embryo with similar regulatory capacity to drive gene activation during the MZT, and may also point to an even more intricate gene regulatory complexity in the embryo.

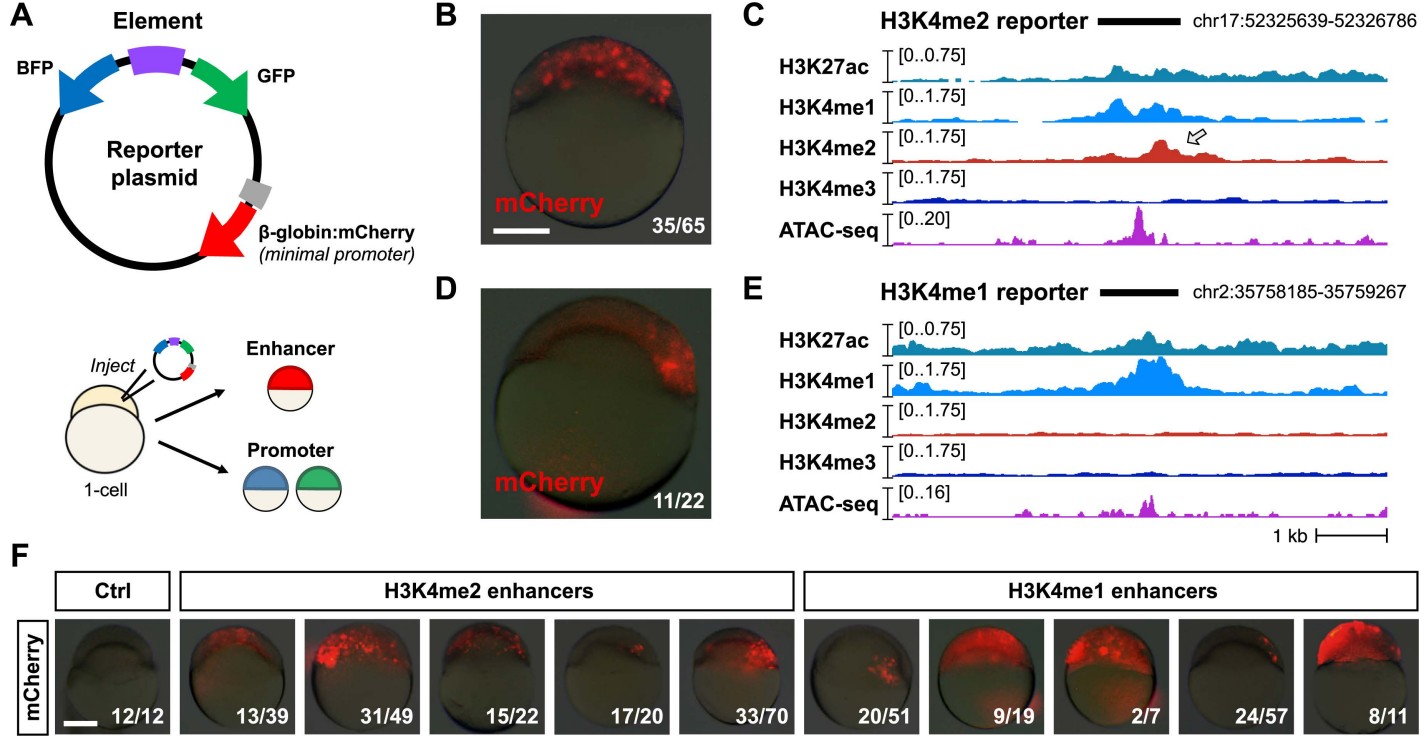

**Fig 3. Reporter assays demonstrate enhancer activity. (A)** Map of the reporter plasmid. Putative regulatory elements are cloned in between divergent mTagBFP2 and EGFP open reading frames to detect (−) strand or (+) strand promoter activity as blue or green fluorescence, respectively. Distal regulation is detected by a far downstream mCherry open reading frame with a minimal mouse β-globin promoter. Reporter plasmids are injected into 1-cell embryos and fluorescence is screened in cells (top of the embryo) in the late blastula / early gastrula. **(B)** mCherry fluorescence from a reporter (Enh_2a) encoding a putative H3K4me2 enhancer. A brightfield image at 25% opacity is overlaid. Fraction of injected embryos fluorescing is shown on the bottom right. **(C)** Genome browser tracks showing CUT&RUN (this study) and ATAC-seq open fragment coverage (data from Liu and colleagues, 2018) over the H3K4me2 reporter tested in **(B)** (black bar). Arrow points to the H3K4me2 enrichment. **(D)** mCherry fluorescence for an H3K4me1 reporter (Enh_1a). **(E)** Genome browser track for the reporter tested in **(D)**. **(F)** mCherry fluorescence for five additional H3K4me2 (Enh_2b-f, middle group) and H3K4me1 enhancers (Enh_1b-f, right group). Control embryos injected with empty reporter plasmids have no fluorescence (left panel). Scale bar = 250 μm.

### H3K4me2 enhancers are activated by maternal mechanisms independent of known pioneer factors

We next sought to understand how H3K4me2 enhancers become active during the MZT. First, to determine whether enhancers gain H3K4 methylation through maternal or zygotic mechanisms, we inhibited genome activation by treating embryos with the Pol II transcription elongation inhibitor triptolide [23,62] and performed CUT&RUN for H3K4me1 and H3K4me2, including a yeast mononucleosome spike-in to aid in normalization (Figs 4A, 6, S4A and S4B). We found that triptolide-treated embryos maintain the pattern of H3K4me1 and H3K4me2 marks observed in DMSO-treated control embryos, again clearly distinguishing the two enhancer classes. Thus, enhancer H3K4 methylation occurs through maternal mechanisms, suggesting that H3K4me2 enhancers can participate in zygotic genome activation.

Next, we asked whether maternal NPS pluripotency factors equivalently regulate both enhancer classes. We analyzed previously published blastula ChIP-seq data for Nanog, Pou5f3, and Sox19b [16,63] and found widespread binding across both H3K4me1 and H3K4me2 enhancers, though with somewhat less intensity for the latter (Fig 4B). When we inspected the underlying sequence for the binding motifs recognized by the factors, we found that H3K4me2 enhancers were significantly depleted for these motifs compared to the H3K4me1 enhancers ($P < 1 \times 10^{-11}$, Chi-squared tests, 2 d.o.f.) (Figs 4B and S6C), suggesting that NPS may not be binding directly or specifically to many of the H3K4me2 enhancers.

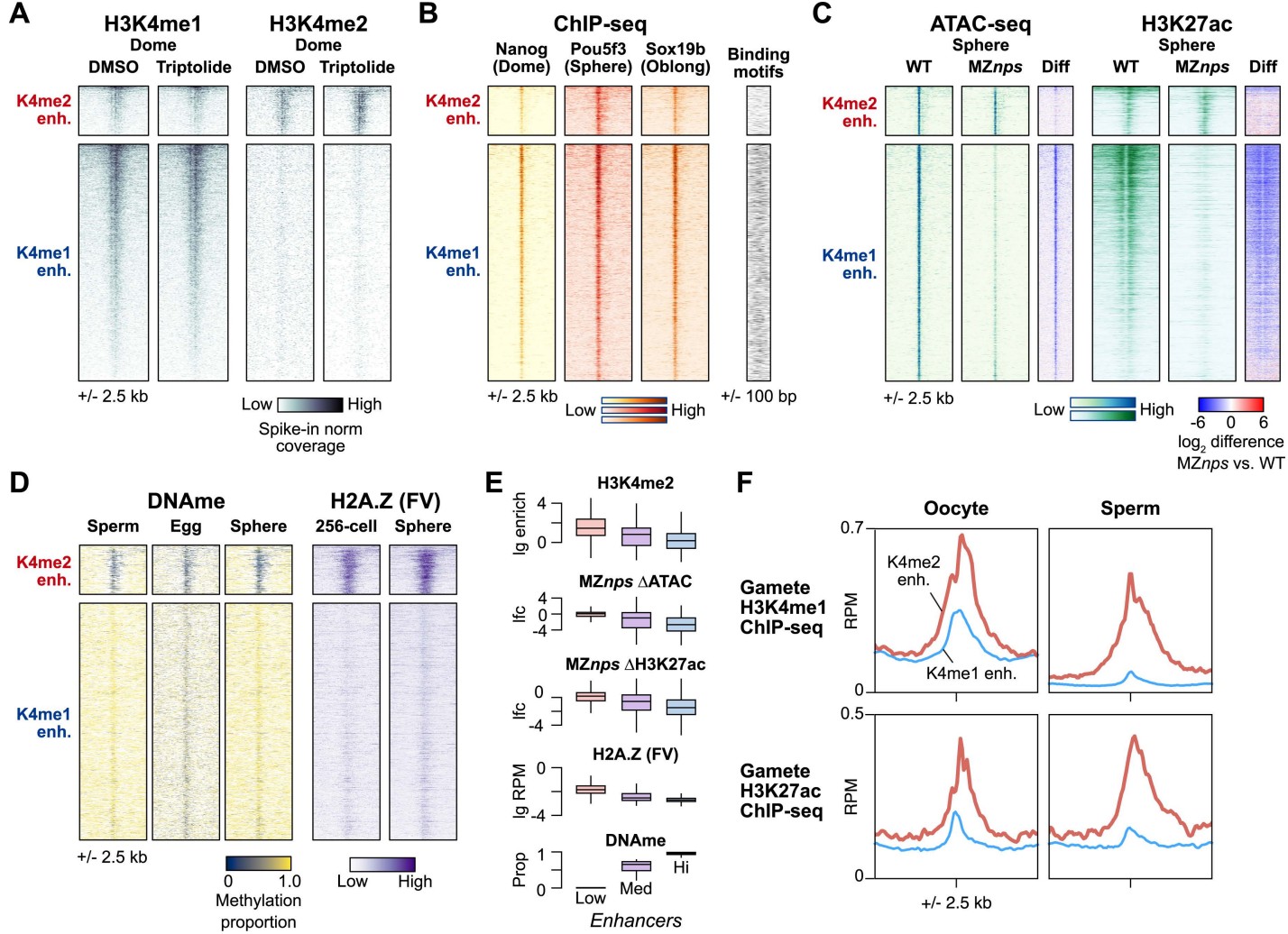

**Fig 4. H3K4me2 enhancers have distinct activation pathways. (A)** Heatmaps over H3K4me2-marked enhancers (K4me2 enh.) and non H3K4me2-marked enhancers (K4me1 enh.) showing replicate-pooled H3K4me1 and H3K4me2 CUT&RUN coverage in control DMSO embryos and embryos treated with the Pol II inhibitor triptolide. Individual replicates are shown in S6A Fig. **(B)** ChIP-seq coverage for Nanog, Pou5f3, and Sox19b (data from Xu and colleagues, 2012 and Miao and colleagues, 2022). Binding motif occurrence for the three factors over the regions is represented as a heatmap on the right. **(C)** ATAC-seq open fragment and H3K27ac ChIP-seq coverage in wild-type embryos and MZ*nps* embryos (data from Miao and colleagues, 2022). Log$_2$-fold difference heatmaps of MZ*nps* coverage vs. wild-type are shown on the right for each chromatin feature. **(D)** DNA methylation proportion from bisulfite sequencing (data from Potok and colleagues, 2013) and H2AFV ChIP-seq coverage (data from Murphy and colleagues, 2018). **(E)** Boxplots comparing correlated chromatin features on enhancers separated into groups with low (<20%), medium (20%–80%), and high (>80%) DNA methylation. Boxes are first through third quartiles, center bar median, whiskers extend to 1.5× the interquartile range, outliers are not shown. The data underlying this figure can be found in S1 Data. **(F)** Aggregate plots for the two embryonic enhancer groups (H3K4me2 enhancers, thick red curves; H3K4me1 enhancers, thin blue curves) showing oocyte and sperm H3K4me1 ChIP-seq average coverage (data from Zhang and colleagues, 2018, and Murphy and colleagues, 2018, respectively) and oocyte and sperm H3K27ac (data from Zhang and colleagues, 2018). The data underlying this figure can be found in S1 Data. lg=log$_2$, lfc=log$_2$ fold change, RPM, reads per million.

In MZ*nps* mutants, the absence of the three maternal pluripotency factors leads to loss of chromatin accessibility and H3K27ac across many enhancers [16]. When we compared sphere-stage ATAC-seq open chromatin and H3K27ac ChIP-seq coverage between wild-type and MZ*nps*, we found that H3K4me2 enhancers indeed do not require NPS for their accessibility or H3K27ac acquisition, in stark contrast to the H3K4me1 enhancers (Fig 4C). Together, these data demonstrate that H3K4me2 enhancers are largely NPS-independent.

PLOS Biology

We considered the possibility that H3K4me2 enhancers may be activated by a yet-unknown maternal transcription factor. However, ChIP-seq binding profiles of other putative maternal activators [16,64–66] showed no strong enrichment at H3K4me2 enhancers over H3K4me1 enhancers (S6D Fig). Motif enrichment analysis revealed some transcription factor binding sequences, but none that seem to unify the H3K4me2 enhancers (S6E and S6F Fig). Thus, NPS pioneering underlies H3K4me1 enhancer activation, but H3K4me2 enhancers as a group seem to activate independent of NPS or any other known sequence-specific pioneer factor.

## H3K4me2 enhancers are hypomethylated and enriched for H2A.Z

In the absence of strong evidence for a novel pioneer factor, we looked instead for epigenetic differences between the two enhancer classes. Previously, Kaaji and colleagues found that putative zebrafish embryonic enhancers exhibit a range of DNA methylation levels, which was also correlated with different chromatin characteristics including H3K4 methylation degree [35], while Murphy and colleagues demonstrated that a subset of hypomethylated embryonic promoters gain accessibility through H2A.Z-containing placeholder nucleosomes [37]. Given that we originally identified H3K4me2 enhancers due to their similarity to promoters, we queried previously published bisulfite sequencing [38] and H2A.Z (H2AFV) ChIP-seq data [37]. We indeed found that H3K4me2 enhancers have an elevated CpG dinucleotide frequency, are strongly hypomethylated in the egg, and maintain low DNA methylation through genome activation (Figs 4D, 4E, S6G and S6H), in contrast to H3K4me1 enhancers, which are hypermethylated. Additionally, H3K4me2 enhancers but not H3K4me1 enhancers acquire strong H2AFV levels (Fig 4D and 4E). Thus, H3K4me2, lack of NPS dependence, low DNA methylation, and H2A.Z are all correlated chromatin features that distinguish a subset of zebrafish embryonic enhancers (Fig 4E).

Genome-wide, DNA methylation patterns in the zebrafish embryo have been found to be reprogrammed to match sperm and not the oocyte/egg [34,38], and indeed we find that here to generally be the case for embryonic enhancers (Figs 4D and S6I). However, a large fraction (69%) of hypomethylated embryonic enhancers is equivalently hypomethylated in both eggs and sperm (S6I and S6J Fig), suggesting that these represent a shared enhancer set used by both gametes and embryos. Indeed, querying existing gamete H3K4me1 and H3K27ac ChIP-seq data [26,37] reveals that the embryonic H3K4me2 enhancers identified here have high levels of these activating histone marks in both oocytes and sperm, while H3K4me1 enhancers do not (Figs 4F and S6K). Thus, H3K4me1 and H3K4me2 enhancers' orthogonal activation pathways may relate to their past activity in gametes: the former rely on maternal factor pioneering to establish de novo activity, while the latter already have a history of activity in gametes and are epigenetically bookmarked to resume activity in the embryo.

## H3K4me2 enhancers likely activate NPS-independent genes

Given that H3K4me2 enhancers are activated through non-NPS dependent pathways, we asked whether they could underlie activation of genes not repressed in MZ*nps* embryos. It is likely that each zygotic gene is regulated by multiple enhancers with variable levels of redundancy, additivity, or synergy that contribute to expression levels, which would complicate deducing regulatory dependence [67]. Despite this, under a strict definition (>3-fold enriched H3K4me2 CUT&RUN signal over IgG), we find that H3K4me2 enhancers are mildly but significantly nearer to non NPS-dependent gene promoters compared to H3K4me1 enhancers ($P = 3.3 \times 10^{-6}$, Wilcoxon rank sum test), indicating a potential regulatory relationship (Fig 5A). This is not the case for NPS-dependent genes ($P = 0.06$, Wilcoxon rank sum test) (Fig 5A). Since we restricted our analysis to putative enhancers ≥2 kb from TSSs, it is possible this trend is underestimated due to the exclusion of TSS-proximal elements. A reexamination of TSS-proximal elements lacking H3K4me3 suggests there may be at least 1,375 additional H3K4me2 enhancers and 441 H3K4me1 enhancers, which echo the properties observed in distal enhancers; though of course, it would be difficult to distinguish these from poised, alternative promoters (S7A and S7B Fig). Including these putative TSS-proximal enhancers strengthens the association between H3K4me2 enhancers and non NPS-dependent genes (S7C Fig).

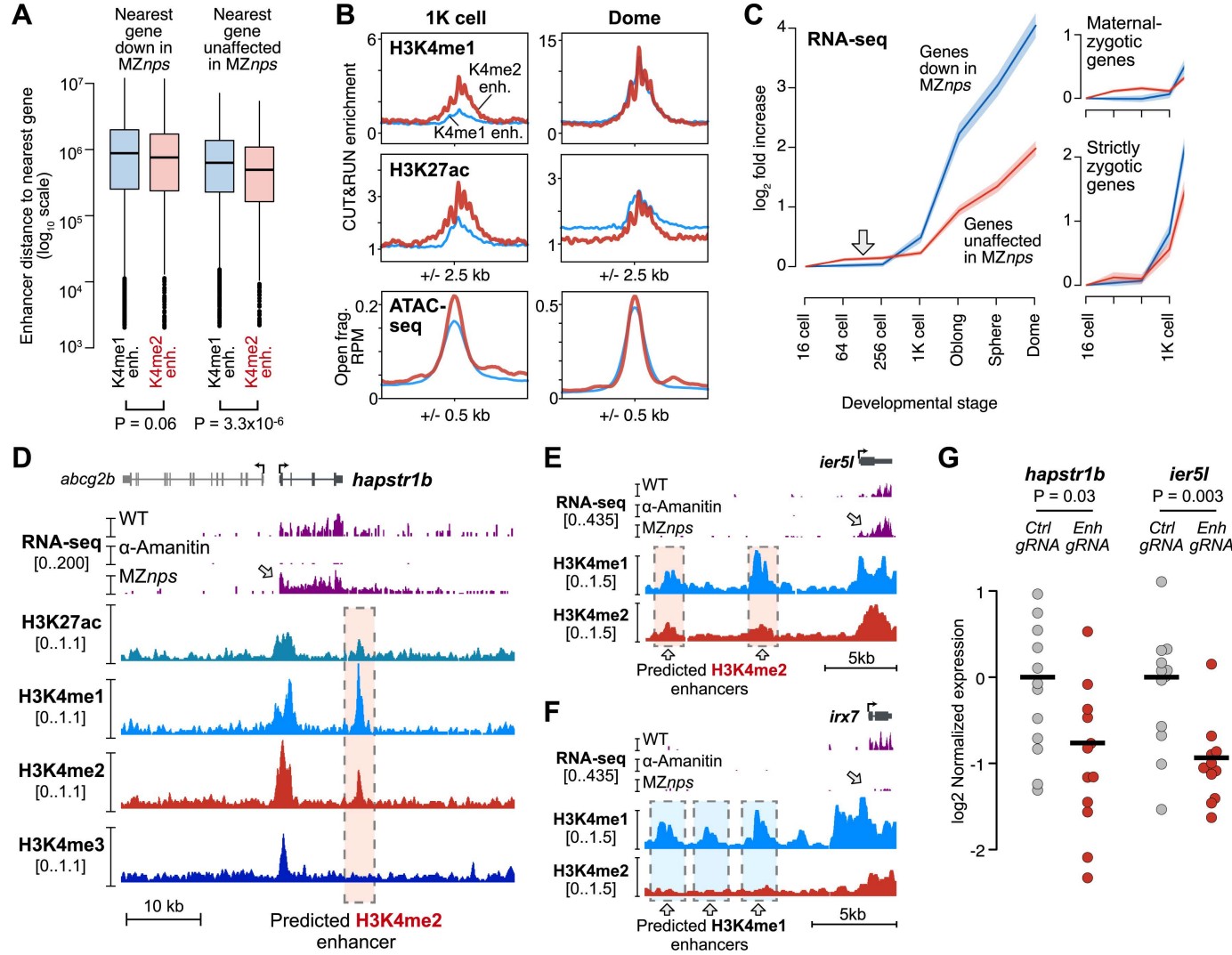

**Fig 5. H3K4me2 enhancers likely regulate non NPS-dependent genes. (A)** Boxplots representing the distance to the nearest gene for each enhancer, for each enhancer/gene combination. Boxes are first through third quartiles, center bar median, whiskers extend to 1.5× the interquartile range, points are outliers. The data underlying this figure can be found in S1 Data. **(B)** Aggregate plots of replicate-pooled CUT&RUN (this study) and ATAC-seq open fragment coverage (data from Liu and colleagues, 2018). H3K4me2 enhancer average plotted as thick red curves, H3K4me1 enhancer average as thin blue curves. The 1K-stage H3K4me1 CUT&RUN shown here uses the Active Motif antibody, dome stage uses the Invitrogen antibody. The data underlying this figure can be found in S1 Data. **(C)** Plot of average wild-type RNA-seq $\log_2$ fold increase over time for genes according to their fate in MZ*nps* embryos – down in MZ*nps* as classified by Miao and colleagues, 2022 (blue line) or unaffected (red line). Ninety-five % confidence intervals are highlighted. Right panels show the plot stratified into genes with a maternal contribution (maternal-zygotic) or strictly zygotic genes. RNA-seq data from Vejnar and colleagues, 2019. The data underlying this figure can be found in S1 Data. **(D–F)** Genome browser tracks illustrating regions with predicted enhancers. Top tracks show Click-iT RNA-seq coverage in wild-type,α-amanitin treated, and MZ*nps* embryos (data from Miao and colleagues, 2022). Lower tracks show CUT&RUN coverage (this study). Predicted enhancers are highlighted with dashed boxes. **(G)** qRT-PCR quantification of zygotic gene expression (*hapstr1b* or *ier5l*) in individual F0 CRISPR-Cas9 enhancer loss-of-function embryos targeting the predicted hapstr1b enhancer shown in **(D)** (left) and two ier5l enhancers simultaneously, shown in **(E)** (right). The data underlying this figure can be found in S1 Data.

Reversing the perspective, NPS-dependent and non-NPS dependent genes seem to have equivalent potential to be regulated by both H3K4me1 and H3K4me2 enhancers, with >95% of genes from either group potentially residing within 1 Mb of either class of enhancers (S7D Fig). This likely reflects the regulatory complexity of promoter-enhancer

relationships, especially given that NPS-dependent genes show varying levels of residual activation even in the absence of NPS [16]. Gene Ontology enrichment analysis comparing genes near each of the two enhancer classes (<100 kb) did not reveal any significantly enriched terms after multiple test correction, though there was a trend for H3K4me2 enhancers to be near genes associated with chromatin structure, and H3K4me1 enhancers to be near genes associated with developmental processes (S7E Fig).

We did however find evidence for a functional connection between H3K4me2 enhancers and non NPS-dependent genes. There is a temporal asymmetry in the activation of the two enhancer classes that is mirrored by the expression dynamics of differentially NPS-dependent genes. At 1K-cell stage, we detect higher levels of H3K4me1, H3K27ac, and chromatin accessibility in H3K4me2 enhancers compared to H3K4me1 enhancers (Figs 2A, 5B and S3A), demonstrating that H3K4me2 enhancers are activated earlier. By dome stage, the signals equalize (Fig 5B, right). Concomitantly, we find that across several RNA-seq time course datasets, non NPS-dependent genes have earlier detectable up-expression than NPS-dependent genes by at least two cell cycles (Figs 5C and S7F–S7I), though NPS-dependent genes subsequently overtake non NPS-dependent genes in magnitude of increase. This phenomenon is unlikely due to dynamic poly(A) tail lengths because we observe the trend in ribosomal RNA-depleted, spike-in normalized datasets (Figs 5C and S7F) as well as with 4SU metabolic labeling of de novo transcription (S7H Fig). The effect seems to be primarily driven by maternal-zygotic gene activation (Figs 5C, right and S7F), consistent with our model where H3K4me2 enhancers are recapitulating oocyte roles during the MZT, reactivating some of the same genes that previously helped shape the maternal contribution (Figs 4F and S6K).

## H3K4me2 enhancer loss of function reduces activation of NPS-independent genes

Finally, we used an F0 CRISPR-Cas9 strategy to target specific H3K4me2 enhancers likely regulating non NPS-dependent zygotic genes (Fig 5D–5F). We injected 1-cell embryos with Cas9 protein complexed with a pool of three different guide RNAs targeting a predicted H3K4me2 enhancer downstream of non NPS-dependent *hapstr1b* (Figs 5D and S8A). We measured *hapstr1b* activation in individual crispant embryos at sphere stage by quantitative reverse-transcription PCR (qRT-PCR) and found on average a 1.7-fold decrease in *hapstr1b* expression compared to control embryos injected with Cas9 + guide RNAs targeting the non-zygotic *slc45a2* (*albino*) promoter ($P = 0.03$, Wilcoxon rank sum test) (Fig 5G). The downregulation is highly variable, as is expected from embryo-to-embryo variability in Cas9 targeting efficacy. As we could not recover sufficient genomic DNA from embryos at such an early developmental stage for genotyping, we instead genotyped sibling crispants at 32 h.p.f. by PCR. We indeed found mosaic patterns of genomic lesions in the *hapstr1b* enhancer locus (S8A and S8B Fig), which likely underlie variable effects on *hapstr1b* activation.

We additionally tested two predicted H3K4me2 enhancers upstream non NPS-dependent *ier5l* (Figs 5E, S5C and S8C), which were not included in our earlier analyses due to lower H3K27ac enrichment at dome stage. Nonetheless, when we targeted both enhancers in parallel with CRISPR-Cas9 and two guide RNAs per enhancer, we found an average 1.9-fold decrease in *ier5l* expression in F0 crispants compared to *albino* controls ($P = 0.003$, Wilcoxon rank sum test) (Fig 5G). Crispant siblings similarly exhibited mosaic genomic lesions (S8C–S8E Fig). Thus, H3K4me2-marked enhancers can regulate zygotic expression of genes that do not depend on maternal NPS pioneer factors.

## Discussion

Here, we have demonstrated that two distinct sets of enhancers regulate the maternal-to-zygotic transition in zebrafish, contributing to widespread gene activation as the embryo induces pluripotent stem cells. Among the 10 histone modifications we profiled using CUT&RUN, it is only H3K4 methylation degree that strongly distinguishes these two enhancer classes. H3K4me3 is not enriched at any enhancer. Putative enhancers marked by H3K4me1 but not H3K4me2 attain chromatin accessibility and activating histone modifications de novo in the embryo through the pioneering activities of maternal pluripotency factors Nanog, Pou5f3, and Sox19b. In contrast, enhancers marked by H3K4me2 are

hypomethylated early, which facilitates acquisition of H2A.Z-bearing nucleosomes that promote open chromatin independent of maternal NPS. A large proportion of these H3K4me2 enhancers overlap with putative hypomethylated oocyte enhancers, suggesting that H3K4me2 enhancers recapitulate gamete regulatory activities in the embryo. Thus, parallel enhancer activation pathways operate during the maternal-to-zygotic transition that are responsible for activating different zygotic gene repertoires (Fig 6).

## A unified model of zygotic genome activation

Our findings unite and extend several previous studies aiming to decipher the regulatory logic of zebrafish embryonic genome activation. The initial discovery that maternally provided pluripotency factors Nanog, Pou5f3, and Sox19b play major roles in genome activation [14,15] reinforced the regulatory connection between transcriptional reprogramming during the maternal-to-zygotic transition in non-mammalian vertebrates and pluripotency induction in mammalian cells. However, these factors did not account for all zygotic gene activation, implicating additional unknown mechanisms. Subsequent elucidation of NPS's pioneering activity at many but not all promoters and enhancers motivated the search for additional factors that could similarly engage and activate nascent, condensed embryonic chromatin [5,16,18,20].

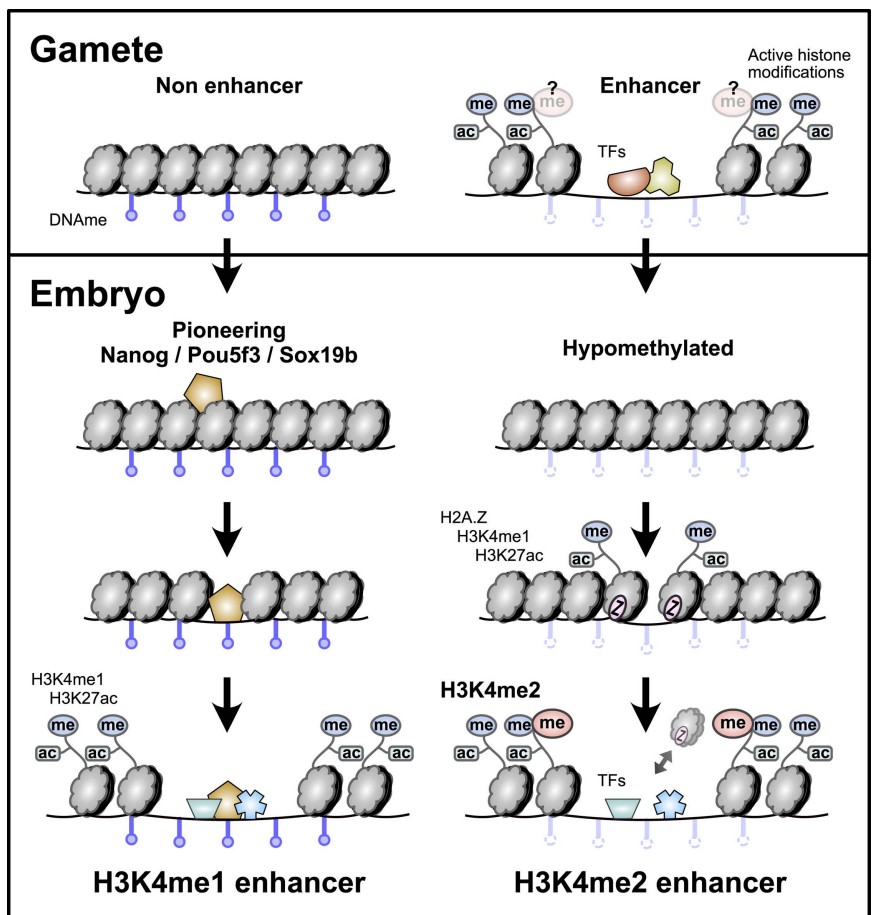

**Fig 6. Parallel enhancer activation pathways during the maternal-to-zygotic transition.** Enhancers that lack evidence for gamete activity are hypermethylated, rely on NPS-pioneering, and are marked with H3K4me1 but not H3K4me2 in the embryo. Enhancers that have evidence for gamete activity are hypomethylated, recruit H2A.Z-containing placeholder nucleosomes rather than relying on NPS pioneering, and are marked with H3K4me2.

Meanwhile, several groups recognized the role of DNA methylation in influencing early embryonic regulatory sequence activity [5,26,33–39]. Hypomethylation was found to be associated with open chromatin at promoters [5,26], and the characterization of embryonic H3K4me1/H2A.Z-bearing placeholder nucleosomes by Murphy and colleagues provided a mechanism for the acquisition and maintenance of promoter accessibility [37]. By contrast, enhancers overall were found to be hypermethylated, which was surprising given the correlation between high DNA methylation and gene repression described in other systems [5,26,35,36]. Zhang and colleagues noted that these enhancers were distinct from gamete enhancers [26], and Liu and colleagues hypothesized that NPS were uniquely capable of binding methylated DNA in the embryo [5]. Kaaij and colleagues recognized that some distal loci were instead hypomethylated, while also bearing bivalent repressive H3K27me3 and activating H3K4me2 and H3K4me3 [35] (though, antibody specificity may have been confounding, see [53]), suggesting that these represented poised enhancers that would later play cell-type-specific roles. Hickey and colleagues subsequently showed that acquisition of repressive H2Aub and later H3K27me3 at hypomethylated enhancers also depended on placeholder nucleosome acquisition [33]. Finally, Wu and colleagues found that inhibiting DNA methylation led to ectopic enhancer activation and acquisition of H3K4me3 (contingent on antibody specificity), further linking hypomethylation with higher order H3K4 methylation [39].

We now find that a subset of hypomethylated enhancers shared with gametes are indeed active in the early embryo and uniquely acquire H3K4me2. These H3K4me2 enhancers likely account for non NPS-dependent embryonic gene activation during the maternal-to-zygotic transition, while enhancers bearing only H3K4me1 correspond to NPS-pioneered enhancers that regulate NPS-dependent genes. This division of labor has implications for how proper transcriptome composition and cellular identity may be maintained throughout germ cell and embryonic development. The maternal contribution is transcribed and curated during the germ cell-to-maternal transition [68,69] to contain the potent reprogramming cocktail centered around Nanog, Pou5f3, and Sox19b, which will eventually induce genome activation and pluripotency in the embryo. Until then, NPS activity presumably must be inhibited to prevent ectopic transcription of developmental triggers. This can be accomplished by limiting their translation until after egg activation [70], but also by inhibiting their target enhancers in the oocyte through DNA methylation. It is still unknown why NPS can activate methylated DNA in the zebrafish embryo, but the high concentration of these factors that accumulates through extremely elevated translation [14] may contribute to their pioneering capacity [71,72]. Conversely, oocyte enhancers that supported transcription of the maternal contribution would not need to be so tightly controlled, since any aberrant activity would simply add to the existing maternal mRNA pool, allowing them to remain poised through hypomethylation to be reactivated in the embryo.

## Clarifying H3K4 methylation degree at enhancers

H3K4 methylation has long been recognized as a hallmark of enhancer loci, and the predominance of H3K4 mono-methylation specifically distinguished enhancers from gene-proximal regions that tend to bear di- and tri-methylation [44–48]. Some reports have suggested that enhancers can indeed attain H3K4me3 [49–52], somewhat blurring the distinction between enhancers and promoters. However, these conclusions are called into question by the recent finding that H3K4 methyl antibody cross-reactivity may contribute to false detection of higher-degree methylation at many loci. Using rigorously tested antibodies, Shah and colleagues demonstrated that at least in K562 cells, only H3K4me1 and H3K4me2 but not H3K4me3 are characteristic of enhancers [53]. Here, we extend these results to zebrafish blastulae: indeed, H3K4me3 is not enriched at enhancers, but we also find that H3K4me2 is not a generic property of all enhancers, but rather marks only a subset of hypomethylated, putative gamete-inherited enhancers that do not depend on pluripotency factor pioneering.

Our findings are reminiscent of a recent report that H3K4me3 marks putative TCF3/12 enhancers in mouse oocytes as well as a subset of enhancers in pre-implantation embryos, during a period of global DNA demethylation [51]. These

enhancers are likely not related to the zebrafish H3K4me2 enhancers, which do not have evidence for TCF3/12 binding (S6E and S6F Fig); and moreover, mammalian ZGA is in many ways mechanistically distinct from zebrafish genome activation [1,2,73,74]. Regardless, together with our findings, this suggests that some higher-order H3K4 methylation at enhancers may be correlated with the transmission of epigenetic information from the germline to the embryo, or during cellular transitions generally, distinguishing persistent or "reawakened" enhancers from "reprogrammed" enhancers that are newly activated. The extent to which this distinction exists in other contexts, e.g., embryonic or artificial pluripotency induction in mammals, remains to be determined.

## Additional regulatory players remain to be elucidated

How precise H3K4 methylation degree is achieved at the two enhancer classes likely involves differential recruitment of chromatin regulators, particularly methyltransferases. Vertebrates encode six major H3K4 methyltransferase variants [75], many of which are duplicated in zebrafish. In vitro, KMT2A/B (MLL1/2) and KMT2F/G (SETD1A/B) are capable of catalyzing all three of mono-, di-, and trimethylation, while KMT2C/D (MLL3/4) can only catalyze mono- and dimethylation [76]. However, the kinetics suggest that KMT2A/B and KMT2C/D preferentially generate H3K4me2 and H3K4me1, respectively [76]. KMT2C/D have been shown to install H3K4me1 at enhancers [77–81], though KMT2A/B have also been found to localize to some enhancers [49,82]. Concordantly, zebrafish H3K4me1 and H3K4me2 enhancers could arise through differential recruitment of these methyltransferases, via mechanisms specific to their respective activation pathways. KMT2A/B contains CXXC domains that direct it to specifically unmethylated CpGs [83–85], which could underlie how hypomethylated enhancers attain H3K4me2. Indeed, Liu and colleagues showed a link between hypomethylated promoter accessibility and Kmt2a and Cxxc1b, the zebrafish ortholog of CXXC1 (CFP1) that complexes with KMT2F/G to similarly target unmethylated CpGs [5,86].

Given that H3K4me2 enhancers in bulk exhibit both H3K4me1 and H3K4me2, KMT2C/D may also target both enhancer classes to deposit H3K4me1, with KMT2A/B subsequently or in competition depositing H3K4me2. Alternatively, exclusive targeting by just KMT2A/B may net either H3K4me1 or H3K4me2 depending on chromatin residency time or genomic context. Thus, embryo-wide variability of the H3K4me2/H3K4me1 ratio at H3K4me2 enhancers may be a function of kinetics over the hundreds to thousands of genome copies and several rounds of rapid replication leading up to genome activation; though, we cannot exclude the possibility of asymmetric H3K4 methylation at the same nucleosome or locus.

We presume that specific maternal transcription factors engage each enhancer and contribute to the recruitment of chromatin factors [16,23]. Although H3K4me1 enhancers seem to be unified by their need for NPS pioneer factor binding, we do not yet know what, if any, transcription factor regulatory logic is shared by H3K4me2 enhancers. Several enriched transcription factor binding motifs are found in various subsets of H3K4me2 enhancer regions (S6E and S6F Fig), so it could be that a diverse collection of factors each bind a subset of different H3K4me2 enhancers. This could account for the dozens of other transcription factors represented in the maternal contribution [14], which likely have combinatorial roles across both enhancer classes in elaborating individual gene expression levels. This is likely true for all enhancers, but it is only a requirement for NPS pioneering that underlies the strong NPS motif signature in H3K4me1 enhancers, a function that is unnecessary for H3K4me2 enhancers. Though, we cannot ignore the transcriptional activating functions of NPS, and indeed at the peak of genome activation, their zygotic gene targets do seem to be more strongly activated on average than non-NPS targets (Figs 5C and S7F–S7I).

Finally, the regulatory logic underlying DNA methylation reprogramming is still incompletely understood. This is particularly relevant for the subset of hypomethylated embryonic enhancers that were previously hypermethylated in the oocyte (S4I and S4J Fig), suggesting that some enhancers may interconvert between activation pathways. Further characterization of the underlying chromatin is warranted as we continue to dissect the regulatory logic of the maternal-to-zygotic transition and embryonic pluripotency induction.

                                                                              

## Methods

### Ethics statement

All animal procedures were conducted under the supervision and approval of the Institutional Animal Care and Use Committee at the University of Pittsburgh, Protocol #21120500.

### Embryo collection

*Danio rerio* were housed in a recirculating aquatic system (Aquaneering) at 28 °C with a 14/10 hr light/dark cycle (light 8 AM to 10 PM). Fish were fed 2× daily (10 AM and 2 PM) with Artemia nauplii. Four to five adult TUAB males and females each were set in divided 1.7 L sloped breeding tanks (Tecniplast #ZB17BTE) overnight. Water was changed and dividers removed at 8–9 AM the following morning, and embryos were collected at 1-cell stage. Embryos were dechorionated by treatment with 1 mg/mL Pronase (Sigma #P5147) in egg water (60 µg/mL ocean salt in DI water) for two minutes then washed. Embryos were incubated at 28.5 °C on agarose coated petri dishes with egg water and collected at appropriate stages as determined by morphology.

For Triptolide (Apexbio #MFCD00210565) treatment, a 4 mM stock solution dissolved in DMSO was added to 1-cell stage embryos in 6-well plates to a final concentration of 2 µM Triptolide and 0.05% DMSO in egg water. DMSO control wells were treated with 0.05% DMSO final. Embryos were collected when DMSO control embryos reached dome stage.

### CUT&RUN

The CUT&RUN procedure was adapted from Hainer and colleagues [55], which incorporates optimizations of the method of Skene and Henikoff [56]. For each sample, approximately 70,000 cells were used: 70 1K-cell stage, 10 dome stage, or 8 shield stage embryos, using average stage cell counts from [57]. Embryos were deyolked in batches of 50–200 embryos: embryos were transferred to 1.5 mL Eppendorf tubes removing excess liquid with a P200 pipettor, then yolk lysis buffer added (55 mM NaCl, 1.8 mM KCl, 1.25 mM NaHCO$_3$). Tubes were shaken at 1,100 RPM for 5 min at room temperature, centrifuged at 300 × $g$ for 30 s to pellet, yolk lysis buffer drawn off, and 1 mL Yolk Lysis Wash Buffer was added (110 mM NaCl, 3.5 mM KCl, 2.7 mM CaCl$_2$, 10 mM Tris pH 8.5). Tubes were shaken at 1,100 RPM for 2 min at RT, centrifuged at 300 × $g$ to pellet, and supernatant was again removed and replaced with 600 µL Nuclear Extraction Buffer (20 mM HEPES-KOH, pH 7.9, 10 mM KCl, 500 µM spermidine, 0.1% Triton X-100, 20% glycerol).

Samples in Nuclear Extraction Buffer were gently resuspended by pipetting up and down, centrifuged at 600 × $g$ at 4 °C for 3 min, supernatant removed, and again resuspended in 600 µL Nuclear Extraction Buffer. To bind nuclei, 150 µL of concanavalin A beads (Polysciences #86057) per sample were activated by added to 850 µL Binding Buffer (20 mM HEPES-KOH pH 7.9, 10 mM KCl, 1 mM CaCl$_2$, 1 mM MnCl$_2$), placed on a magnet stand, and washed twice with Binding Buffer. Beads were resuspended in 300 µL Binding Buffer and slowly added to nuclei with gentle vortexing (approximately 1,500 rpm), then rotated 10 min at RT. Supernatant was drawn off on a magnet stand, then beads were blocked for 5 min in 1 mL Wash Buffer (20 mM HEPES-KOH pH 7.5, 150 mM NaCl, 0.5 mM spermidine, 0.1% BSA w/v) with 2 mM EDTA for 5 min at RT. To bind antibody, supernatant was drawn off on a magnet stand and washed 2× with 1 mL Wash Buffer. Beads were resuspended in 500 µL of 1:100 primary antibody in Wash Buffer for 2 hr at 4 °C on a rotator. To bind pAG-MNase, beads were washed 2× in 1 mL Wash Buffer, then resuspended in 500 µL of 1:200 pAG-MNase (gift from Sarah Hainer) in Wash Buffer for 1 hr at 4 °C, and washed again 2× with Wash Buffer. Beads were resuspended in 150 µL Wash Buffer and placed on ice for 5 min, then the pAG-MNase was activated by adding 3 µL 100 mM CaCl$_2$ while gentle vortexing and returning to ice. After 30 min, the reaction was stopped using 2× STOP Buffer (200 mM NaCl, 20 mM EDTA, 4 mM EGTA, 50 µg/mL RNase A, 40 µg/mL glycogen; and 10 pg/mL yeast mononucleosome as a spike-in (20 pg/mL for the Triptolide experiments). Nuclei were incubated at 37 °C for 20 min followed by centrifuging for 5 min at 16,000 × $g$ at 4 °C, drawing off the DNA fragments with the supernatant. The extracted fragments were treated with SDS (0.1%) and

proteinase K (2.5 μL of 20 mg/mL stock) at 70 °C for 10 min followed by phenol chloroform extraction and ethanol precipitation. Purified DNA was resuspended in 50 μL of water. Antibodies used were: H3K4me1, Invitrogen #710795, lot #2477086 (all stages), and ActiveMotif #39297, lot #01518002 (for 1K-cell stage only); H3K4me2, Invitrogen #710796, lot #2246656; H3K4me3, Invitrogen #711958, lot #2253580; H3K27ac, Abcam #ab4729, lot #GR3357415-1; H3K9ac, Cell Signaling #9649, lot #13; H3K56ac, Invitrogen #PA5-40101, lot #XA3485152A; H3K64ac, Abcam #ab214808, lot #GR3312057-4; H3K122ac, Abcam #ab33309, lot #GR3427528-1; H4K16ac, Millipore #37707329, lot #3770263; H2BK16ac, Abcam #ab177427, lot #GR199432-1; IgG, Invitrogen #10500C. CUT&RUN libraries were constructed using the NEB Ultra II DNA library prep kit (NEB #E7645) and indexed adapters according to manufacturer's protocol. DNA was end repaired and then ligated to sequencing adaptors diluted 1:100. Ligated DNA was purified with 0.9× Sera-Mag Select beads (Cytiva #29343045) and PCR amplified for 14-15 cycles, then purified again with 0.9× Sera-Mag beads. Libraries were run on a 1.5% TBE agarose gel, and a band corresponding to 175–650 bp was cut out and gel purified using the NEB Monarch DNA gel extraction kit (#T1020). Concentration was verified by Qubit dsDNA high sensitivity and Fragment Analyzer. Sequencing libraries were multiplexed and paired-end sequenced on an Illumina NextSeq 500 at the Health Sciences Sequencing Core at Children's Hospital of Pittsburgh.

## In vivo reporter assay

For the enhancer reporter plasmid, starting with a pTol2 α-crystallin mCherry plasmid, CMV:EGFP was amplified from pCS2+ cytoplasmic EGFP (gift from Antonio Giraldez) using F-aaactagagattcttgtttagaattcGTCGACCATAGCCAAT-TCAATATGGC and R-ctagagtcgaGGTACCGGGCCCAATGCA and inserted using NEB HiFi Assembly (NEB #E5520). The β-globin minimal promoter was amplified from mouse genomic DNA (gift from Sarah Hainer) using F-aaaggta-CCAATCTGCTCAGAGAGGACA, R-aaagctagcGATGTCTGTTTCTGAGGTTGCA and cloned into the plasmid with KpnI/NheI to replace the existing mCherry promoter. mTagBFP2 was amplified from pBS mTagBFP2 (derived from pCS2+ mTagBFP2-LL2, gift from Carson Stuckenholz) with F-aactagagattcttgtttaGGAACAAAAGCTGGAGCTCCACC, R-tgaattggctatggtcgacgAATTCCTGCAGCCCGGGG and inserted using HiFi Assembly. To flip the CMV promoter (to generate the CMV:BFP version), the plasmid was cut with BamHI (flanks both sides of CMV) and re-ligated. Candidate regulatory regions were amplified from genomic DNA (approximately 800–1,200 bp) and cloned into the plasmid cut with EcoRI/HindIII using HiFi assembly or classical cloning. Primers are listed in S3 Table. Sequences were verified by whole plasmid sequencing (Plasmidsaurus) and concentrations quantified by Qubit.

An amount of 30 pg of each reporter plasmid was injected into dechorionated 1-cell embryos into the cell using a PV 820 Pneumatic Pico Pump. Fluorescence was visualized at 6 h.p.f. on a Leica M165 FC scope with a FlexCam C3 camera with the following settings: Gamma: 1.5, Sharpness: 10, Noise Reduction: 4, Saturation: 0. For each fluorophore, settings were: mCherry, Exposure 125 ms, Gain 35 dB; BFP: Exposure 125 ms, Gain 28 dB; GFP: Exposure 88.3 ms, Gain 22 dB. Images were edited in Adobe Photoshop using the Levels function, setting the output levels to be (Shadows/Gamma/Highlights): mCherry 30/1/55, GFP 22/1.13/122, BFP 69/0.67/109. BFP was false colored to cyan by changing Hue under Hue/Saturation to −50. Grayscale images were generated by inverting each corrected image, then setting threshold to 0% for all colors in the Black and White setting. For the high-sensitivity insets, the Levels lower threshold (Shadows) was set to 10 for mCherry, 7 for GFP, and 33 for BFP; then inverted to black and white as before. Brightfield images were auto white balanced on the scope, then in Photoshop, Color Balance for Highlights was adjusted up to −50 on the Magenta/Green scale.

## CRISPR mutagenesis

Cas9 crRNAs were designed referring to the IDT design tool and CRISPRscan [87] and synthesized by IDT (Alt-R-XT for *albino* and *ier5l*, Alt-R for *hapstr1b*) and resuspended to 100 μM in IDT duplex buffer. crRNAs were hybridized with tracrRNA (IDT) and complexed with Cas9 protein (Alt-R S.p Cas9 Nuclease 3NLS, IDT #1074181) as described in [88]:

                                                    

final concentration 10 µM Cas9, 10 µM gRNA duplex (equimolar pool of multiple guides), 0.04% Phenol red in a 5 µL volume. CRISPR crRNA sequences: *hapstr1b (*ENSDARG00000012458) enhancer: GGTGACATTGTACTGAGTGG, TGTTAGCTGCTGACCCCTAG, TCTTTGATGAGAAATGAGCG. *ier5l* (ENSDARG00000054906) proximal enhancer: TCCGGTGGCAGGAGGACCAG, ACAACAGTAGGCTACCATGG. *ier5l* distal enhancer: TGCGCGCTGCAGGGTGACAG, CGTGGAAGTGTTAGCAGCAC. *slc45a2* (ENSDARG00000002593, *albino*) promoter (negative control): TCAAGACTTGT-GAGCTGAGA, TCCTGCTGGGAGTGGACAAT. Guides were pooled per gene for each set of injections (i.e., all three *hapstr1b* guide were pooled, all four *ier5l* guides were pooled).

Dechorionated 1-cell embryos were injected with 1 nL Cas9 complex into the cell. Embryos were incubated at 28.5 °C and a portion were collected at sphere stage, individually flash frozen in liquid nitrogen, and RNA was extracted by TRIzol (Invitrogen #15596026), quantified by NanoDrop, and stored at −80 °C until use for qRT-PCR. Sibling embryos were collected at 32 hrs post-fertilization for genotyping: individual embryos were boiled at 95 °C in 100 mM NaOH for 20 min, followed by neutralization with 1 M Tris-HCl (pH 7.4) and stored at −20 °C until use.

For qRT-PCR, 40 ng RNA per embryo was used as template for the Luna Universal One-Step RT-qPCR Kit (NEB #E3005S) with three technical replicates per embryo per primer pair. qRT-PCR was carried out on a QuantStudio 3 96-Well 0.1 mL Block machine with the following cycling conditions: an initial 10 min incubation at 55 °C, followed by 40 cycles of 95 °C, 10 s; 60 °C, one minute. Ramp speed was 1.6 °C/s. Ct values for technical replicates were averaged, then per embryo Ct values for the target gene (*hapstr1b* or *ier5l*) were normalized to the reference gene (*dusp6* ENS-DARG00000070914, an NPS-dependent zygotic gene to control for ZGA timing) (ΔCt). Values were converted to $2^{-\Delta Ct}$ and then normalized by the control embryo average so that the control embryo average value was 1 (0 on a log scale) for graphing. Primers were: *dusp6*: F-AGCCATCAGCTTTATTGATGAG and R-CAAAGTCCAAGAGTTGACCC (209 bp exon 2–3), *hapstr1b*: F-TGTGTGTGTTATTTGAACGGGA and R-TAGGTTAGTGACGGCAGTTG (158 bp exon 2+ intron, nascent transcript), *ier5l*: F-TGCAGTGGATGCACAAAGTC and R-ATCTCCGCGTACTTCTCGTT (156 bp, single-exon gene).

For genotyping, 1 µL of template was used in a 25 µL PCR reaction (NEB 2× Ultra Q5 master mix, #M0544S), Ta = 67 °C, 30 s ext., and run on a 2% TAE gel. Genotyping for *ier5l* enhancer deletion involved amplifying each enhancer locus separately, then in another reaction amplifying the region spanning both enhancers using the distal forward and proximal reverse primers to detect large deletions. Primers were: *hapstr1b* enhancer: F-TTCAGCACACATTTCTTTTCTGT, R-AGACAGCCTTCAACAATA CACA, *ier5l* distal enhancer: F-CCATTGGATTCGTGACGCAC, R-TACTTGCGTGCCTACTCCTC, *ier5l* proximal enhancer: F-TCGTGGGTTATTCTTTTACGCC, R-TTGAAGTGTGTTTTGCGTTGC.

## Data analysis

**CUT&RUN alignment.** Paired-end reads were mapped to the zebrafish genome (GRCz11) using bowtie2 v2.4.2 [89] (--no-mixed --no-discordant -X 650). Filtered FASTQ files for each CUT&RUN library were first assembled by removing contaminating read pairs that align the hg38 human genome and not the zebrafish genome (GRCz11) (total read pair counts reported in S1 Table). High-quality alignments to zebrafish (MAPQ ≥ 30) were retained, after additional filtering to also exclude reads mapping chrM, or to satellite DNA or rRNA as annotated by RepeatMasker. For the PCA analysis, only mononucleosome-sized CUT&RUN fragments (140–250 bp spanned by read pair) were used, which were trimmed (tag-centered) to 73 bp, then filtered to exclude duplicate regions with identical start/end coordinates (filtered read pair counts reported in S1 Table). To normalize triptolide CUT&RUN samples with yeast spike in, unaligned reads were aligned to the sacCer3 genome to obtain the number of total unique yeast read pairs, and BigWigs were scaled by 1e6/yeast pairs. Downstream analyses were performed using Linux shell scripts with the aid of UCSC Genome Browser – Kent tools [90], BEDtools v2.30.0 [91], Samtools v1.12 [92], and deepTools v3.5.1 [93].

**ATAC-seq analysis.** Accessible regions were defined using ATAC-seq datasets from Liu and colleagues, GEO: GSE101779 [5] and Pálfy and colleagues, GEO: GSE130944 [18] (All public datasets used are listed in S4 Table). For the

Liu and colleagues dataset, reads from 1K-cell, oblong, and dome stages were aligned to GRCz11 using bowtie2 (--no-mixed --no-discordant --dovetail -X 2000), retaining read pairs with MAPQ > 2 with fragment length < 120 bp. Reads were clipped using Trim Galore (-e 0.2) [94] prior to mapping. Peaks were called on the union of the stages using Macs2 [95] with an effective genome size of 4.59e8 (GRCz11 summed chromosome length minus sum RepeatMasker annotated regions). For the Pálfy dataset, published accessible regions were lifted over from GRCz10 to GRCz11, then regions not overlapping the Liu peaks were added to the analysis. In total, there were $N = 41,334$ accessible regions from the Liu dataset (each region named in the form atac_L00001, atac_L00002, …, atac_L41334) and $N = 7,256$ additional regions from the Pálfy dataset (atac_P00001, …, atac_P07256), for a grand total of $N = 48,590$ regions.

**Promoter/enhancer annotation.** To identify promoters, published CAGE-Seq data from dome, shield, and 14-somite stages, SRA: SRP013950 [96] was used in conjunction with Ensembl r100 gene annotations to select the maximally zygotically expressed TSS per Ensembl gene (supported by >20 cage tags). Additional TSSs for genes not annotated by Ensembl were added from RefSeq and UMMS v4.3.2 [58] annotations. ATAC-seq accessible regions that overlap a TSS were annotated as a promoter ($N = 10,299$). To classify enhancers, all remaining ATAC-seq accessible regions <2 kb from any annotated TSS transcript isoform were classified as TSS-proximal elements (alternate promoters or proximal enhancers, $N = 11,899$), while regions ≥2 kb from any annotated TSS were classified as distal elements, i.e., enhancers ($N = 26,197$). ATAC-seq open regions on unassembled scaffolds lacking annotated genes were discarded, leaving $N = 48,395$ total regions.

**PCA and enhancer class definitions.** Raw CUT&RUN read coverage was calculated over the promoter and distal regions using bedtools coverage in the ATAC-seq open interval, 500 bp upstream the interval, and 500 bp downstream the interval (3 counts per ATAC region, per CUT&RUN sample). Sixty-three intervals lacking 500 bp of flanking sequence (e.g., on the edge of a scaffold) were discarded. For dome-stage PCA, dome stage CUT&RUN coverage counts were used, keeping replicates as separate features for the first PCA, and pooling replicates per histone mark for the second PCA. IgG samples were not included. Counts were normalized by region length as $\log_2$ RPKM/2 (i.e., per 500 bp rather than per 1 kb), adding a pseudocount of 1. PCA was performed using R 4.1.0 prcomp with input matrix of 48,332 regions × 66 features for the unpooled PCA and 30 features for the pooled PCA (upstream, center, downstream per histone mark; "downstream" was set to be the flanking region with higher total CUT&RUN coverage summed over all marks). For the time-course PCA with all marks, coverage for all non-IgG, unpooled samples across 1K-cell, dome, and shield stages was used as independent features (135 features total). For the timecourse H3K4me PCA, each of the 48,332 ATAC-seq regions was represented by 3 points (rows) corresponding to CUT&RUN coverage at each of 1K-cell, dome, and shield stages. The features consisted of H3K4me1, H3K4me2, and H3K4me3 coverage upstream, center, and downstream of the ATAC-seq peak. This yielded an input matrix of 144,996 rows × 9 features. CUT&RUN coverage counts were pooled across replicates (all 1K-cell stage H3K4me1 samples across both antibodies were pooled).

To define the enhancers that cluster with promoters (H3K4me2 enhancers) and promoters that cluster with enhancers (Low K4me2/3 promoters), rotated data for the first 3 PCs of the pooled PCA were input into an SVM classifier using the *R* svm function in the e1071 package v1.7-13 using gamma = 1, cost = 1; only Ensembl promoters and distal enhancers were used. The SVM model was used to classify all regions using the predict function. For contour lines on the biplot visualizations (Fig 1E), a 2D density kernel estimation was calculated for the first 2 PCs using the R kde2d function in the MASS package v7.3-54, $h = 3$, $n = 125$. For initial heatmap visualization (Figs 1 and 2), regions with ≥2-fold dome-stage H3K4me1 enrichment over IgG and ≥10 RPKM coverage in the center + downstream interval were used, $N = 4,128$ H3K4me1 enhancers, $N = 644$ H3K4me2 enhancers, $N = 4,707$ High H3K4me2/3 promoters, $N = 1,224$ Low H3K4me2/3 promoters. For subsequent analyses, refined active enhancer categories were used: ≥1.5-fold pooled dome-stage H3K27ac enrichment over IgG; and ≥2-fold pooled dome-stage H3K4me2 for H3K4me2 enhancers, with an SVM classification of "promoter"; or ≥2-fold pooled dome-stage H3K4me1 and <1.25-fold H3K4me2 enrichment with an SVM classification of "enhancer" for H3K4me1 enhancers. Additional H3K4me2 enhancers were rescued from the SVM "enhancer" class if they

met the H3K4me2 enrichment thresholds and also had high H3K4me2 coverage (equivalent to the top 25th percentile of bona fide H3K4me2 enhancers). Poised enhancers with ≤1.5-fold H3K27ac enrichment were also annotated for reference. A subset of TSS-proximal elements were also annotated as potential enhancers if they had <1.25-fold H3K4me3 enrichment, and ≥2-fold H3K4me2 enrichment for potential H3K4me2 enhancers.

**CUT&RUN heatmaps.** CUT&RUN coverage heatmaps were generated using deepTools computeMatrix reference-point (--referencePoint center -b 2,500 -a 2,500 --binSize 25 --missingDataAsZero) [93] with adaptive color scales per histone mark: zMin in the plotHeatmap command is set to the mean upstream signal in the leftmost 20 25-bp bins as calculated by computeMatrix, zMax is set to the 90th percentile of the signal in the center 8 bins. CUT&RUN enrichment heatmaps over IgG were plotted using fold-difference bigWigs generated by bigwigCompare (--operation ratio --skipZeroOverZero --pseudocount 0.1 --binSize 50); plotHeatmap colors ranged from zMin 1 (i.e., no enrichment) to zMax 10 for H3K4me1/2, zMax 4 for other marks. All heatmaps are uniformly sorted relative to descending H3K4me1 signal unless otherwise indicated.

**RNA-seq coverage over enhancers.** To assess RNA-seq signal at putative enhancers, strand-specific RNA-seq coverage was calculated in a 100 bp window upstream and downstream (relative to genomic coordinates) of each ATAC-seq open interval, using poly(A)+ RNA-seq datasets at dome, 50% epiboly, shield, and 75% epiboly stages [60]. Potential (+)-strand gene TSSs were regions with ≥1 RPKM (+)-strand coverage downstream that is ≥2-fold higher than (+)-strand coverage upstream, in at least two time points; (−) strand, ≥1 RPKM (−)-strand coverage upstream ≥2-fold higher than downstream. ATAC-seq open regions whose 100 bp flanks fall within a known annotated exon were not considered potential TSSs (i.e., RNA-seq signal is likely due to the surrounding gene). Patterns were confirmed with a second RNA-seq dataset of the same stages from [61].

**Other chromatin and DNA methylation dataset comparisons.** ChIP-seq coverage heatmaps were generated using deepTools computeMatrix reference-point (--referencePoint center -b 2,500 -a 2,500 --binSize 25 --missingDataAsZero) [93]. Wild-type versus MZ*nps* chromatin heatmaps were generated using deepTools bigwigCompare (--operation log2 --skipZeroOverZero --pseudocount 0.01). DNA methylation was visualized by processing previously published bisulfite sequencing, SRA:SRP020008 [38] using bwa-meth [97] and MethylDackel extract (--mergeContext --minDepth 10) (github.com/dpryan79/methyldackel). Heatmaps for methylation proportion were generated using computeMatrix --binSize 100 and omitting the --missingDataAsZero parameter, and plotHeatmap using --interpolationMethod nearest to improve aesthetics. For chromatin feature boxplots (MZ*nps* log fold change, H2AZ coverage, DNA methylation proportion), average signal over the central 500 bp for histone features or 200 bp for DNA methylation and ATAC-seq was extracted from the deepTools computeMatrix data tables used for heatmap generation.

**Motif enrichment analysis.** NPS motif density was calculated in a ±100 bp window centered on the ATAC-seq open interval using the homer2 find command [98] on empirically determined Nanog, Pou5f3, and Sox19b motifs from performing homer2 de novo motif finding on zebrafish ChIP-seq [16]. A bigWig of motif hit coordinates/occurrences was used as input to deepTools for visualization. Motif scanning in enhancer groups was performed using homer2 findMotifs.pl on 200 bp of sequence centered on ATAC open intervals. H3K4me2 enhancer sequences were used as foreground and H3K4me1 enhancer sequences were used as background, then for a separate analysis each enhancer group was used as foreground with background sequences consisting of non-exonic ATAC-seq open regions with <1.25-fold dome-stage CUT&RUN enrichment for any histone mark ($N$=2,132). For TOBIAS 0.17.1 motif analysis [99], ATACcorrect was run on the merged ATAC-seq BAM file and MACS peaks from above, with a blacklist of satellite DNA and rRNA as annotated by RepeatMasker; then ScoreBigwig was run using a BED file of H3K4me1 enhancers only, and again using H3K4me2 enhancers only. Finally, BINDetect was run comparing the H3K4me1 and H3K4me2 footprinting files across the merged enhancer set, using the JASPAR 2024 CORE vertebrates non redundant PFMs [100]. TOBIAS ClusterMotifs (--threshold 0.2) was used to group motifs by similarity, such that only the most significant motif per cluster is shown on the volcano plot. For base composition analysis, CpG and C+G content was calculated in the center 500 bp of each element using bedtools nuc.

**Gene ontology enrichment analysis.** GO terms for all sphere-stage zygotic activated genes as defined by [16] were obtained from the DAVID Functional Annotation web tool [101]. GO term annotation frequency was calculated for genes within 100 kb of H3K4me1 enhancers and genes within 100 kb of H3K4me2 enhancers and enrichment was assessed by a Fisher's exact test, followed by FDR multiple test correction.

**Definitions of NPS-down versus NPS-unaffected genes.** Annotations were obtained from [16] ($N = 691$ down, $N = 1,100$ unaffected). Enhancer distances to each gene group were calculated using bedtools closest, discarding enhancers on unassembled scaffolds. RNA-seq time-course trajectories were calculated for each gene group as the mean $\log_2$ expression at each time point − $\log_2$ expression at time 0, using a pseudocount of 0.1. Ninety-five % confidence bounds per time point were calculated as $\pm qt *$ standard deviation/sqrt($n$) where $n =$ the number of genes and $qt$ is the 0.975 quantile of the $t$ distribution with $n − 1$ degrees of freedom. Published normalized expression values from each study were used and joined with the Miao and colleagues gene IDs, with some genes dropping out due to different annotations used between studies. For the Vejnar and colleagues dataset [60], yeast spike-in normalized unique counts from rRNA-depleted RNA-seq were used. Maternal-zygotic genes were defined as having pooled 2-cell expression at >0.5 RPKM.

## Supporting information

**S1 Fig. Principal component analysis on histone modifications. (A)** Biplot showing the first two principal components (PCs) from the full dome-stage CUT&RUN principal component analysis (PCA) (replicates not pooled). Percent of total variance explained per PC in parentheses. Points represent ATAC-seq accessible regions and are colored by proximity to annotated TSSs: promoters (orange) overlap TSSs, promoter-proximal elements (green) are within 2 kb of a TSS, and distal elements (enhancers, blue) are at least 2 kb away from any annotated TSS. Each of the three groups is plotted individually to the right, showing all outliers. **(B)** As in **(A)**, but showing PC1 versus PC3. **(C)** Heatmap of the loadings from the PCA showing the contribution of each feature to the first three PCs. Each replicate contributes three features: CUT&RUN coverage upstream, within, and downstream of the ATAC-seq open region, as defined in Fig 1B. **(D)** Full biplot of the first two principal components (PCs) for the pooled PCA as in Fig 1D, including outliers far from the main point masses. Percent of total variance explained per PC in parentheses. Points are labeled as enhancers (blue) or promoters (orange). **(E)** Heatmap of the loadings from the PCA. Columns are principal components, rows are input features – histone modification coverage on upstream, center, and downstream regions of predicted regulatory elements. **(F)** Biplots as in **(A)** for PCs 3–6. The data underlying all panels of this figure can be found in S1 Data.
(EPS)

**S2 Fig. Dome-stage histone modifications. (A)** Heatmaps of CUT&RUN coverage as in Fig 1F showing individual replicates. **(B)** Strand-specific sphere-stage nascent RNA-seq coverage over the two promoter classes, High H3K4me2/3 and Low H3K4me2/3. Heatmaps are centered on the TSS with ±0.5 kb context. RPKM = Reads per kilobase per million transcripts. **(C)** Heatmaps of regions stratified by the fourth principal component (PC4) of the pooled PCA, which H3K56ac strongly contributes to. H3K56ac has been proposed to be an Oct4-associated enhancer mark in mouse ES cells (Tan and colleagues, 2013). PC4 high = ATAC-seq regions with PC4 values greater than the 1 standard deviation from 0, PC4 low = less than −1 * standard deviation. Enhancers stratified by PC4 have differential H3K56ac, but do not differ in other enhancer-associated marks. Parallel heatmaps of Nanog, Pou5f3 (Oct4 homolog), and Sox19b ChIP-seq coverage (data from Miao and colleagues, 2022) demonstrate no correlation with PC4/ H3K56ac. No differences are seen among promoters either. Thus, H3K56ac likely does not define a distinct regulatory element class in zebrafish embryos. **(D)** Heatmaps of regions enriched for H3K122ac (>2-fold over IgG), stratified by H3K27ac co-enrichment (<1.25-fold or >1.5-fold), showing very few regions marked by H3K122ac without H3K27ac. Pradeepa and colleagues (2016) identified a subclass of H3K122ac+/H3K64ac+/H3K27ac− enhancers in mouse, but enhancers with that pattern do not seem to be prevalent in zebrafish embryos.
(EPS)

**S3 Fig. Genomic profiles over time. (A)** Heatmaps of CUT&RUN coverage for histone modifications at 1K-cell stage centered on enhancer and promoter regions as defined in Fig 1F. Individual replicates are shown. Two different H3K4me1 antibodies were used, Active Motif #39297 and Invitrogen #710795 (the same antibody used for all other time points). **(B)** Heatmaps of CUT&RUN coverage at shield stage showing individual replicates. **(C)** Principal component analysis (PCA) parameterized as in S1A Fig but now including all CUT&RUN replicates across all three developmental stages. Percent of total variance explained per PC in parentheses. Points are colored according to membership in the four categories defined in Fig 1F: H3K4me1 enhancers (blue), H3K4me2 enhancers (red), High H3K4me2/3 promoters (orange), Low H3K4me2/3 promoters (aqua). Each of the groups is plotted individually below. For visual consistency with the PCA plots in S1A Fig, the PC2 axis is flipped. **(D)** Plots of PCs 1−3 for an H3K4 methylation CUT&RUN PCA, using pooled-replicate CUT&RUN for each stage; 1K-cell stage H3K4me1 pools together samples from both the Active Motif and Invitrogen antibodies. PC2 and 3 are the *x*- and *y*-axes, respectively, PC1 is represented by a color scale, where dark blue indicates high H3K4 methylation and yellow indicates low (note that the numerical scale is −1 * PC1). The ATAC-seq open regions for the four element categories (columns) are plotted over time (rows), showing movement through PCA space as their H3K4 methylation profiles change. **(E)** Average PCA trajectories over time for each of the four element categories. H3K4me1 enhancers start near the origin at 1K-cell stage, then move southeast to quadrant IV as they activate (arrows point to the path between stages), while H3K4me2 enhancers move north to quadrant I over time (top panel), reflecting their different H3K4 methylation profiles. High H3K4me2/3 promoters move northwest to quadrant II, while Low H3K4me2/3 promoters only begin to move north to intersect the High H3K4me2/3 path at shield stage, suggesting they have delayed activation. The data underlying panels **C–E** can be found in S1 Data.
(EPS)

**S4 Fig. Assessing transcriptional regulation at regulatory elements (A) Heatmaps of ChIP-seq for repressive histone modifications.** Each heatmap is sorted by descending signal per region group independently. Data are from Zhu and colleagues, 2019 (1K-cell H3K27me3), Zhang and colleagues, 2014 (dome H3K27me3), Duval and colleagues, 2024 (H3K9me3), and Hickey and colleagues, 2022 (H2Aub). **(B)** Strand-separated RNA-seq coverage heatmaps as in Fig 2B showing intermediate developmental stages. Data are from Vejnar and colleagues, 2019 (TUAB strain) and White and colleagues, 2017 (HLF strain).
(EPS)

**S5 Fig. Reporter assays for regulatory elements. (A)** Representative embryos injected with reporter plasmids (top to bottom, CMV promoter oriented toward the mTagBFP2, CMV promoter oriented toward the EGFP, and H3K4me2 reporter Enh_2a) imaged in brightfield and BFP, GFP, and mCherry channels. Corresponding grayscale images are shown in (A′). Fraction of injected embryos fluorescing is shown on the bottom right of each panel (counts for all other reporters are listed in S3 Table). **(B, C, D)** Representative embryo fluorescence for additional mCherry-positive enhancers. Color brightfield and mCherry panels are shown for each reporter, along with grayscale panels for mCherry, GFP, and BFP. bicra_prox is an element proximal (<2 kb) to the *bicra* TSS. Enhancers Enh_2i and Enh_2j are the *ier5l* enhancers tested in the CRISPR-Cas9 experiments. **(E, F)** Representative embryos for promoter reporters showing color brightfield images and grayscale mCherry, GFP, BFP channels. Insets show the respective channels after lowering the pixel display threshold 3× to show faint signal. **(G)** RNA-seq expression of the *arhgap18* gene over development, from 1-cell stage to 5 days post fertilization. Data are from White and colleagues, 2017. The *x*-axis is labeled with intervals corresponding to activities during the maternal-to-zygotic transition (cytoplasmic polyadenylation, which yields increased levels in poly(A)+ selected RNA-seq without actual transcript number increases; and maternal mRNA clearance, leading to degradation of maternally contributed mRNA) and beyond. The estimated stage of *arhgap18* activation is labeled (1–4 somites stage). The data underlying this panel can be found in S1 Data. TPM = transcripts per million. Scale bar = 250 μm.
(EPS)

**S6 Fig. Chromatin characteristics of the two enhancer classes. (A)** CUT&RUN heatmaps as in Fig 4A showing individual replicates. Each replicate is a paired DMSO control and triptolide treatment group. **(B)** Representative images of time-matched control DMSO-treated embryos and Triptolide-treated embryos. Triptolide treatment inhibits transcription, which leads to failed epiboly. The asterisk indicates an unfertilized egg. **(C)** Barplots showing proportion of enhancers containing predicted zebrafish Nanog, Pou5f3, and Sox19b binding sequences as represented by sequence logos (left) empirically determined from ChIP-seq (data from Miao and colleagues, 2022). $P$ values for Chi-squared tests (1 d.o.f.) are listed on the right. **(D)** Heatmaps showing ChIP-seq coverage for different embryonic transcription factors on enhancer regions as well as 1,000 ATAC-seq open regions lacking enrichment for any dome-stage histone modifications. Each heatmap is individually sorted in descending order per group. Boxplots summarizing coverage are below each heatmap (boxes are first through third quartiles, center bar median, whiskers extend to 1.5× the interquartile range, outliers are not shown). Data are from Dubrulle and colleagues, 2015 (dome stage FoxH1 and Smad2), Miao and colleagues, 2022 (sphere stage Nfya and Eomesa), Ladam and colleagues, 2018 (high/oblong stage Prep1), Stanney and colleagues, 2020 (high/oblong-stage Pbx4). **(E)** Volcano plot from TOBIAS motif-enrichment analysis comparing H3K4me1 enhancers to H3K4me2 enhancers. Only one hit per transcription-factor cluster is shown. **(F)** Table of top enriched transcription factor binding motifs in H3K4me2 enhancers relative to H3K4me1 enhancers from Homer analysis. One representative motif per family is shown. **(G, H)** Boxplots showing CG dinucleotide (CpG) and C+G nucleotide prevalence in 500 bp centered on H3K4me1 enhancers, H3K4me2 enhancers, and active TSSs. $P$-values for Wilcoxon rank sum tests are shown. **(I)** Biplots comparing DNA methylation proportion in gametes ($x$ axis) versus sphere stage embryos ($y$ axis) for predicted enhancers. **(J)** Stacked barplot showing the proportion of hypomethylated (<20% methylated) embryonic enhancers that are also hypomethylated in gametes. **(K)** Gamete H3K4me1 and H3K27ac ChIP-seq heatmaps over embryonic enhancers. Data from Murphy and colleagues, 2018 (sperm H3K4me1) and Zhang and colleagues, 2018. The data underlying panels **C–I** can be found in S1 Data.
(EPS)

**S7 Fig. Enhancer association with zygotic genes. (A)** Expanded enhancer annotations: similar to Fig 4, heatmaps of chromatin features over TSS-proximal elements excluded from the main enhancer analysis (<2 kb from, but not overlapping, any TSS, regardless if there is evidence for zygotic expression). Elements with H3K4me3 enrichment (top) are considered to be alternate promoters. The remaining elements segregate into possible H3K4me2 enhancers (middle group) and possible H3K4me1 enhancers (bottom group). Reporter assays suggest that such promoter-proximal regions can function as enhancers, despite the ambiguity in annotating them as such (S3D Fig). **(B)** Boxplots similar to Fig 4E summarizing the correlated chromatin features for TSS-proximal possible enhancers, which likewise segregate into hypermethylated, NPS-dependent, exclusively H3K4me1-marked enhancers and hypomethylated, non NPS-dependent, H3K4me2-marked enhancers. **(C)** Boxplots similar to Fig 5A, but including TSS-proximal elements. **(D)** Bar plots showing proportion of genes with 100 kb (top) or 1 Mb (bottom) of H3K4me1 enhancers (blue bars) and H3K4me2 enhancers (pink bars). Darker shaded region of each bar represents proportions limited to only strictly defined enhancers (TSS distal and <1.25-fold or ≥3-fold enriched for H3K4me2 for H3K4me1 and H3K4me2 enhancers, respectively). Lighter shaded regions include TSS-proximal elements. **(E)** Gene ontology enrichment analysis comparing GO terms between activated genes 100 kb from H3K4me1 enhancers versus H3K4me2 enhancers. Terms with positive log enrichment are enriched in genes near H3K4me1 enhancers, negative log enrichment are enriched in genes near H3K4me2 enhancers. Uncorrected $P$ values from Fisher's tests are shown. None of the enrichments are significant after FDR multiple test correction. **(F–I)** Plots of average wild-type RNA-seq $\log_2$ fold increase over time as in Fig 5C. MZ = maternal-zygotic genes only, Z = strictly zygotic genes only. Data are from Vejnar and colleagues (2019), White and colleagues (2017), Heyn and colleagues (2014), and Bhat and colleagues (2023). The data underlying panels **B–I** can be found in S1 Data.
(EPS)

**S8 Fig. Genotyping F0 enhancer crispants. (A)** *hapstr1b* locus showing the ATAC-seq open region, CRISPR guide RNA target sites, and genotyping primers at the predicted downstream enhancer region. **(B)** Genotyping gel for single embryos at 32 h post fertilization. Lane 1 = NEB 1 kb Plus ladder, lane 2 = wild-type, lanes 3–12 = embryos injected with a pool of Cas9 complexed with *hapstr1b* enhancer guide RNA. **(C)** *ier5l* locus showing two upstream predicted enhancers annotated as in **(A)**. **(D)** Genotyping gels for single embryos using primers to detect lesions in the proximal enhancer (top) and distal enhancer (bottom). Crispants were injected with a pool for all gRNAs targeting both enhancers. Gel configuration similar to **(B)**. **(E)** Genotyping gels for the same embryos as in **(D)** to detect large deletions spanning the two *ier5l* enhancers. The wild-type product (6,559 bp) should not efficiently amplify under the PCR conditions used. Bands appearing in the wild-type lane are likely off-target products (asterisk). Uncropped gel images can be found in S1 File. (EPS)

**S1 Table. CUT&RUN samples generated in this study.**
(XLSX)

**S2 Table. Regulatory regions defined in this study.**
(XLSX)

**S3 Table. Enhancer reporters.**
(XLSX)

**S4 Table. Sources of public data used.**
(XLSX)

**S1 Data. Data underlying panels of** Figs 1, 4, 5, **S1**, **S3**, **S5**, **S6**, and **S7**.
(XLSX)

**S1 File. Raw images. Uncropped gel images.**
(PDF)

## Acknowledgments

We thank S. Hainer for providing the pAG-MNase enzyme and C. Stuckenholz for the tagBFP2 plasmid; S. Hainer, K. Arndt, A. Carlson, M. Rebeiz, M. Tsang, C. Kaplan and their labs for equipment use and feedback. This project used the University of Pittsburgh Health Sciences Core at UPMC Children's Hospital Pittsburgh for sequencing.

## Author contributions

**Conceptualization:** Matthew D. Hurton, Miler T. Lee.

**Data curation:** Miler T. Lee.

**Formal analysis:** Matthew D. Hurton, Miler T. Lee.

**Funding acquisition:** Miler T. Lee.

**Investigation:** Matthew D. Hurton, Jennifer M. Miller, Miler T. Lee.

**Methodology:** Matthew D. Hurton, Miler T. Lee.

**Project administration:** Miler T. Lee.

**Supervision:** Miler T. Lee.

**Validation:** Matthew D. Hurton, Miler T. Lee.

**Visualization:** Matthew D. Hurton, Miler T. Lee.

**Writing – original draft:** Matthew D. Hurton, Miler T. Lee.

**Writing – review & editing:** Matthew D. Hurton, Miler T. Lee.

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
