## [Editor Report · Decision Letter 0]

Dear Dr Lee,

Thank you for submitting your manuscript entitled "H3K4me2 distinguishes a distinct class of enhancers during the maternal-to-zygotic transition" for consideration as a Research Article by PLOS Biology.

Your manuscript has now been evaluated by the PLOS Biology editorial staff as well as by an academic editor with relevant expertise and I am writing to let you know that we would like to send your submission out for external peer review.

Once your full submission is complete, your paper will undergo a series of checks in preparation for peer review. After your manuscript has passed the checks it will be sent out for review. To provide the metadata for your submission, please Login to Editorial Manager (https://www.editorialmanager.com/pbiology) within two working days, i.e. by Sep 12 2024 11:59PM.

Kind regards,

Ines

--

Ines Alvarez-Garcia, PhD

Senior Editor

PLOS Biology

---

## [Decision Letter · Decision Letter 1]

Dear Dr Lee,

Thank you for your patience while your manuscript entitled "H3K4me2 distinguishes a distinct class of enhancers during the maternal-to-zygotic transition" was peer-reviewed at PLOS Biology. Please also accept my apologies for the delay in providing you with our decision. The manuscript has now been evaluated by the PLOS Biology editors, an Academic Editor with relevant expertise, and by two independent reviewers.

The reviews are attached below. As you will see, both reviewers find the conclusions potentially interesting, however they also raise several issues that would need to be addressed before we consider the manuscript for publication. Reviewer 1 has several suggestions for improvement, including the analysis of how long the H3K4me2 enhancers persist during development. Reviewer 2 requests a more comprehensive PCA analysis genome-wide across different replicates and histone modification to see how these factors influence the data. In addition, this reviewer thinks you should perform a footprinting analysis using TOBIAS to further differentiate between two classes of enhancers, among other experiments.

In light of the reviews, which you will find at the end of this email, we would like to invite you to revise the work to thoroughly address the reviewers' reports. Given the extent of revision needed, we cannot make a decision about publication until we have seen the revised manuscript and your response to the reviewers' comments. Your revised manuscript is likely to be sent for further evaluation by all or a subset of the reviewers.

**IMPORTANT - SUBMITTING YOUR REVISION**

3. Resubmission Checklist

a) *PLOS Data Policy*

b) *Published Peer Review*

Sincerely,

Ines

--

Ines Alvarez-Garcia, PhD

Senior Editor

PLOS Biology

Reviewers' comments

Rev. 1:

The manuscript "H3K4me2 distinguishes a distinct class of enhancers during the maternal-to-zygotic transition" by Hurton et al. explores mechanisms for the initial activation of embryonic genes that are not regulated by the canonical trio of zebrafish pluripotency factors. To do this, they employ CUT&RUN to rigorously profile the histone tail landscape of zebrafish chromatin in blastula stage embryos around the time of the onset of the major initial wave of zygotic transcription. Intriguing, they find two subclasses of enhancers that are distinguished by the presence or absence of H3K4me2. Enhancers lacking H3K4me2 are associated with activation by canonical pluripotency factors, while those with H3K4me2 appear to be hypomethylated and to gain accessibility independent of these factors. The manuscript is thorough and the data sets that are generated are likely to provide an important resource for a wide variety of investigators. Arguments for reevaluating H3K4 methylation are well articulated, and the resulting observation that there are potentially two unique sets of enhancers marked by different H3K4me valencies reveals a new aspect of the mechanisms governing transcriptional activation that will be of interest to researchers working in areas of gene regulation, the maternal-to-zygotic transition, and early cell fate decisions. The fact that accessibility is governed by distinct mechanisms is of interest even though the transcriptional relationships between H3K4me2 enhancers and pluripotency factor independent gene expression were not especially clear cut.

Suggestions for manuscript improvement include:

Line 50. Minor, but I'm not sure the supplied references provide evidence that during the first two hours post-fertilization, chromatin is tightly condensed. Perhaps " has limited accessibility as assessed by ATAC seq" or something of that nature might be more precise?

Can the authors provide additional information on aligned reads and percent aligned reads in Sup table 1?

It would be helpful for the authors to state the number of H3K4me2 marked enhancers, it seems to be a relatively small subset.

In figure one and throughout the paper, it could be useful to indicate when data presented are merged data across replicates and the number of replicates.

In figure 1. It is not clear to this reviewer what comparisons this statement refers to "H3K4me2 through H2BNTac have significant differences each at P < 1x10-100, and the remaining marks are significant to P < 1x10-30" is there a way to make this clearer?

In figure 1, it seems enhancers marked by H3K4me2 are also marked by H3K4me1, can the authors be more clear on if they are suggesting these two marks occur together on adjacent nucleosomes? Is it possible embryos are heterogeneous for the two states, with these regions marked by me1 in some cells and me2 in others

It could be useful to understand how long these H3K4me2 enhancers persist during development, the title alludes to them in the context of MZT, but might they be more broadly relevant during extended embryogenesis? Their persistence at shield stage seems to raise this possiblity.

The authors use RNA seq data from white et al to help distinguish promoter and enhancer regions. However, I believe they use a different strain background compared to white et al which could complicate this analysis. Could there be cases where different TSS are used between the two stains in the early embryo? It might be prudent to acknowledge this caviate

Can the authors provide evidence that their triptolide treatment worked?

Can the developmental stage for the presented data be included in the legend for experiments in Figure 4?

Can the authors provide more clarity on how methylation proportion was calculated in figure 4? (it could be helpful to break up the data analysis section in the method, it is quite long and dense, making it hard to pull out details, apologies if I missed this) Do K4me2 enhancers show similar CpG density to K4me1 enhancers?

Rev. 2:

H3K4me2 distinguishes a distinct class of enhancers during the maternal-to-zygotic transition

General thoughts:

This work explores the mechanisms driving embryonic genome activation in zebrafish during the maternal-to-zygotic transition. While vertebrate pluripotency factors Nanog, Pou5f3, and Sox19b (NPS) are known to pioneer chromatin opening and enable gene activation, some genes are transcribed without these factors, indicating the presence of alternative regulatory mechanisms. Using CUT&RUN to profile histone modifications, the authors identify two types of enhancers: those marked by H3K4me2, which are epigenetically bookmarked and function independently of NPS, and those lacking H3K4me2, which depend on NPS for activation. These findings reveal dual pathways for enhancer activation in early embryonic development.

Overall, the paper presents an interesting model distinguishing two classes of enhancers and leverages multiple experimental modalities to explore this concept. At certain points it does lack an overall view of the data created for the analysis. It can also benefit from some more in depth analysis, like using the staged ATAC-seq to explore these regions and potentially GO-enrichment analysis on the genes near the two different classes to look at if there is any biological trend to NPS-independent regulated genes. The enhancer reporter assay could also be expanded, with examples of promoter regions from the two sub-populations.

Major comments:

Figure 1 & Supplementary Figure 1:

* It would be beneficial to include a more comprehensive PCA analysis genome-wide across different replicates and histone modifications. This would provide a clearer picture of how these factors influence the data and allow for a more robust interpretation of the results.

* The classification of certain promoters as "enhancer-like" is not entirely convincing. It is unclear if these are genuinely enhancer-like or simply inactive promoters. A more thorough analysis, perhaps utilizing ATAC-seq data across developmental stages (as shown in Liu et al., 2018), could clarify whether these promoters are dynamically regulated.

* Figure 1SE is currently not well explained in the results section and appears somewhat disconnected from the overall message. Providing more context and a clearer rationale in both the figure legend and the results would make it easier for readers to grasp its relevance.

Figure 2 & Supplementary Figure 2:

* Like the suggestion for Figure 1, a PCA analysis for these histone marks could provide additional insights, particularly regarding whether developmental progression is captured.

* There seems to be a lack of signal in the data for the Invitrogen antibody used in H3K4me1, which raises the question of whether Invitrogen data was used for analysis.

Figure 3 & Supplementary Figure 3:

* I have a general concern regarding the classification of enhancers as "promoter-like" in Figure 1 based on PCA data. From the limited sampling, there does not appear to be a strong indication that "promoter-like" enhancers have a higher potential to act as promoters than typical enhancers. A more detailed justification or additional data would be helpful.

* Including examples of typical promoters and "enhancer-like" promoters would be very informative. Are these promoters inactive in general, or do they exhibit enhancer-like activity? This comparison could help clarify the functional differences between these two categories.

* Figure S3D is quite confusing and is not adequately discussed in the results section. Additionally, it seems to imply that promoters would produce similar signals as enhancers, raising questions about whether this assay is suitable for distinguishing enhancer activity from promoter activity.

* It would also be helpful to display the signals for all colors in the panels.

Figure 4 & Supplementary Figure 4:

* To further differentiate between these two classes of enhancers, performing a footprinting analysis using TOBIAS would be a valuable addition. Since ATAC-seq data is already available at the relevant developmental stages, this analysis could provide additional insights into differential potential transcription factor binding events and enhancer function across the two classes of enhancers.

Minor comments:

Figure 1 & Supplementary Figure 1:

o The data presented in panel A could more clearly indicate that it is from the dome stage to clarify the specific developmental stage being analyzed.

o The introduction of terms such as "promoter-like enhancers," "typical promoters/enhancers," and "enhancer-like promoters" can be confusing. It would be easier on the reader to use a consistent terminology throughout the paper. For instance, the histone modification-based naming convention (e.g., "H3K4me1 enhancers" and "H3K4me2 enhancers") could be introduced early on and used consistently.

Figure 3 & Supplementary Figure 3:

o The legend for Fig. 3S (panel A) is missing a reference to the relevant Supplementary Table number.

o The BFP signal is hard to see. Using a more distinct color overlay (e.g., yellow) would enhance visibility and make the data more interpretable.

Figure 4 & Supplementary Figure 4:

o The discussion in lines 219-220, stating that no motifs unify H3K4me2, is somewhat vague. Comparing panel SD to panel SB, it doesn't appear that motifs like Nanog, Pou5f3, or Sox19b unify H3K4me1 more strongly than H3K4me2..

o In panel F, some of the text is cut off.

Figure 5 & Supplementary Figure 5:

o It might be better to call open regions using macs3 to make use of hmmratac for potentially more robust analysis.

---

## [Decision Letter · Decision Letter 2]

Dear Dr Lee,

Thank you for your patience while we considered your revised manuscript entitled "H3K4me2 distinguishes a distinct class of enhancers during the maternal-to-zygotic transition" for publication as a Research Article at PLOS Biology. This revised version of your manuscript has been evaluated by the PLOS Biology editors, the Academic Editor and one of the original reviewers.

Based on the reviews, we are likely to accept this manuscript for publication, provided you satisfactorily address the data and other policy-related requests stated below my signature.

We expect to receive your revised manuscript within two weeks.

*Published Peer Review History*

*Press*

Sincerely,

Ines

--

Ines Alvarez-Garcia, PhD

Senior Editor

PLOS Biology

DATA POLICY:

Fig. 1D, G; Fig. 2E, F; Fig. 3A-C, G; Fig. S1A-E; Fig. S3C-E; Fig. S6C-E, G, H and Fig. S7B-I

**Please note that we cannot accept sole deposition of data in GitHub, as this could be changed after publication. However, you can archive this version of your publicly available GitHub code to Zenodo. Once you do this, it will generate a DOI number, which you will need to provide in the Data Accessibility Statement (you are welcome to also provide the GitHub access information). See the process for doing this here: https://docs.github.com/en/repositories/archiving-a-github-repository/referencing-and-citing-content

CODE POLICY

Reviewers' comments

Rev. 2:

The authors have satisfactory addressed our questions and we do not have any further corncers about the work.

---

## [Editor Report · Decision Letter 3]

Dear Dr Lee,

Thank you for the submission of your revised Research Article entitled "H3K4me2 distinguishes a distinct class of enhancers during the maternal-to-zygotic transition" for publication in PLOS Biology. On behalf of my colleagues and the Academic Editor, Carmen Williams, I am delighted to let you know that we can in principle accept your manuscript for publication, provided you address any remaining formatting and reporting issues. These will be detailed in an email you should receive within 2-3 business days from our colleagues in the journal operations team; no action is required from you until then. Please note that we will not be able to formally accept your manuscript and schedule it for publication until you have completed any requested changes.

PRESS

Sincerely, 

Ines

--

Ines Alvarez-Garcia, PhD

Senior Editor

PLOS Biology
